# Land surface parameter optimisation using data assimilation techniques: the adJULES system

Nina M. Raoult[1], Tim E. Jupp[1], Peter M. Cox[1], and Catherine M. Luke[1]

[1]National Centre for Earth Observation, University of Exeter, Exeter EX4 4QF, UK

*Correspondence to:* Nina Raoult (nr278@exeter.ac.uk)

**Abstract.** Land-surface models (LSMs) are crucial components of the Earth System Models (ESMs) which are used to make coupled climate-carbon cycle projections for the 21st century. The Joint UK Land Environment Simulator (JULES) is the land-surface model used in the climate and weather forecast models of the UK Met Office. JULES is also extensively used offline as a land-surface impacts tool, forced with climatologies into the future. In this study, JULES is automatically differentiated with respect to JULES parameters using commercial software from FastOpt, resulting in an analytical gradient, or adjoint, of the model. Using this adjoint, the adJULES parameter estimation system has been developed, to search for locally optimum parameters by calibrating against observations. This paper describes adJULES in a data assimilation framework and demonstrates its ability to improve the model-data fit using eddy covariance measurements of gross primary production (GPP) and latent heat (LE) fluxes. adJULES also has the ability to calibrate over multiple sites simultaneously. This feature is used to define new optimised parameter values for the 5 Plant Functional Types (PFTs) in JULES. The optimised PFT-specific parameters improve the performance of JULES at over 85% of the sites used in the study, at both the calibration and validation stages. The new improved parameters for JULES are presented along with the associated uncertainties for each parameter.

## 1 Introduction

Land-surface models (LSMs) have formed an important component of climate models for many decades now (Pitman, 2003). First generation land-surface schemes focussed on providing the lower boundary condition for atmospheric models by calculating the land-atmosphere fluxes of heat, moisture and momentum, and updating the surface state variables on which these fluxes depend (e.g. soil temperature, soil moisture, snow-cover). In the mid to late 1990s some land-surface modelling groups began to introduce additional aspects of biology into their schemes, most notably the dynamic control of transpiration by leaf stomata and the connected rates of leaf photosynthesis (Sellers et al. (1997); Cox et al. (1999)).

In the early 2000s, climate modelling groups began to use the carbon fluxes simulated by LSMs within first generation climate-carbon cycle models (Cox et al. (2000), Friedlingstein et al. (2001)). These early results, and a subsequent model inter-comparison (Friedlingstein et al., 2006), highlighted the uncertainties associated with land carbon-climate feedbacks. The 5th Assessment Report of the Intergovernmental Panel on Climate Change (IPCC AR5 (Stocker et al., 2013)) for the first time routinely included models with an interactive carbon cycle (now called Earth System Models or ESMs), confirming that land responses to climate and $CO_2$ are amongst the largest of the uncertainties in future climate change projections (Arora and Boer

(2005); Brovkin et al. (2013); Jones et al. (2013); Friedlingstein et al. (2013)). Any future decreased ability of the land surface to draw-down atmospheric $CO_2$ could imply smaller "compatible emissions" in order to stay below key warming thresholds such as two degrees.

Uncertainties in LSMs arise from three major sources: parameter uncertainty, process uncertainty and uncertainty due to initial and boundary conditions. Taking these in reverse order, uncertainty due to initial and boundary conditions refers to uncertainty in the forcing data (Kavetski et al. (2006a, b), Ajami et al. (2007)). Process uncertainty includes the misrepresentation of land-surface processes and also the neglect of important processes (such as nitrogen-limitations on plant growth, see for example Thornton et al. (2007); Zaehle et al. (2010)), or canopy light interception (Mercado et al., 2009). The drive to

reduce process uncertainty almost invariably leads to increases in LSM complexity, which typically leads to the introduction of additional internal model parameters. Parameter uncertainty arises from uncertainty in these internal model parameters. The evolution of LSMs has therefore involved an attempt to reduce process uncertainty by increasing model realism/complexity, but at the cost of increasing parameter uncertainty. This paper concerns the development and application of a technique to reduce parameter uncertainty in the widely used Joint UK Land Environment Simulator (JULES) LSM (Best et al. (2011); Clark et al.

(2011)).

    Optimisation techniques come under the umbrella of model-data fusion and range from simple ad-hoc parameter tuning to rigorous data assimilation frameworks. These approaches have been used in a number of studies, covering various LSMs, to derive vectors of parameters that improve model-data fit significantly (e.g. Wang et al. (2001, 2007); Reichstein et al. (2003); Knorr and Kattge (2005); Raupach et al. (2005); Santaren et al. (2007); Thum et al. (2008); Williams et al. (2009); Peng et al.

(2011)). Many of these studies calibrate the model at individual measurement sites. Given the small spatial footprint of each flux tower, this can often result in over-tuning. This over-tuning may occur when a single site does not represent the full range of a Plant Functional Type (PFT), given different tree types, tree ages and above ground biomass found at each site. There may be some anomalous plants in the small footprint that are not representative of the PFTs over a broader area. The optimised model parameters are site-specific and often struggle to perform as well when generalised over other sites (Xiao et al., 2011).

The majority of LSMs group vegetation into a small number of PFTs. Model parameters are assumed to be generic over each PFT. Through different optimisation techniques, some studies have tried to assess the robustness of PFT-specific parameters (e.g. (Kuppel et al., 2014)). Medvigy et al. (2009) and Verbeeck et al. (2011) both show that parameters derived at one site can perform well on a similar site and over the surrounding region (Medvigy and Moorcroft (2011)). However, a contradictory study by Groenendijk et al. (2010) found that there was cross-site parameter variability after optimisation within the PFT

groupings.

    In the last few years, there has been a move towards deriving PFT-specific parameters using data from multiple sites, the results of which have been generally positive (e.g. Xiao et al. (2011) and Kuppel et al. (2012)). Both of these studies used data from multiple sites in their optimisation (calling it multisite optimisation) and have commented on the robustness of this technique, showing that the choice of the initial parameter vector had little effect on the optimised values.

    Kuppel et al. (2012) compared different approaches for finding generic PFT-specific parameters, such as averaging optimised parameter vectors over PFTs and directly optimising over multiple sites. They found that the latter method was best for finding

PFT-specific parameters. The multisite optimisation procedure was refined in Kuppel et al. (2014), extended to other PFTs, and evaluated at a global scale.

For global modelling, there is a clear need to find generic parameters and associated uncertainties for each PFT, by optimising against observations in a reproducible way. This paper presents a model-data fusion framework, called adJULES, that allows data from multiple sites to be used simultaneously in order to improve the JULES land surface model. The adJULES system uses the adjoint method which finds minima rapidly across multiple parameters via matrix inversion and has the advantage of reproducibility. Replicating these findings using brute-force optimisation would be prohibitively expensive computationally.

This paper aims to answer the following questions:

- Can a (locally) optimum vector of generic parameters for each of the JULES PFT classes be found?

- How does the optimal PFT parameter vector compare to parameter vectors found by optimising each site individually?

- How robust is the adJULES system when optimising over multiple sites?

- What uncertainty is associated with each parameter?

In section 2, methods and data used in the study are described. The JULES land surface model and our new data assimilation system (adJULES), are introduced, along with the data used and the parameters chosen to be optimised in the study. In section 3, the results are presented. The methodology for optimising over multiple sites simultaneously is validated, and optimum parameter values are provided for each JULES PFT. The performance of the new parameter sets is assessed and shown to improve significantly the fit of the JULES model to the observations. The conclusions are laid out in section 4.

## 2  Methods and Data

### 2.1   The JULES land-surface model

The JULES land-surface model (Best et al., 2011; Clark et al., 2011) simulates the interactions between the land and the atmosphere. Originally developed from the Met Office Surface Exchange Scheme (MOSES) (Cox et al., 1999), JULES can be used 'offline' with observed atmospheric forcing data, or can be coupled into a global circulation model (GCM). JULES is the
land surface model used in the UK Met Office Unified Model.

JULES is a mechanistic land surface model including physical, biophysical, and biochemical processes that control the radiation, heat, water, and carbon fluxes in response to time-series of the state of the overlying atmosphere (Best et al., 2011; Clark et al., 2011). Processes such as evaporation, plant growth and soil microbial activity are all linked through mathematical equations that quantify how environmental conditions affect evapotranspiration, heat balance, respiration, photosynthesis and
carbon assimilation (Best et al., 2011; Clark et al., 2011). JULES runs at a given sub-daily step (typically 30 minutes), using meteorological drivers of rainfall, incoming radiation, temperature, humidity, and windspeed as inputs.

Vegetation in the JULES model is categorised into five PFTs; broadleaf trees (BT), needleleaf trees (NT), C3 grasses (C3G), C4 grasses (C4G), and shrubs (Sh). Default parameters for these PFT classes are taken from a previous study (Blyth et al., 2010).

The eight parameters that are calibrated within this study (see Table 1) relate predominantly to leaf-level stomatal conductance ($g$) and photosynthesis ($A$). Four of these parameters control the responses of $g$ and $A$ to environmental conditions such as surface temperature ($T_{upp}$, $T_{low}$), solar radiation ($\alpha$), and atmospheric humidity deficit ($dq_c$). Two other calibration parameters ($f_0$, $n_{l0}$) essentially control the maximum values of $g$ and $A$. The remaining two calibration parameters influence the hydrological partitioning at the land-surface and relate to the amount of rainfall intercepted by the plant canopy ($\delta c/\delta L$),

and the "rootdepth" ($d_r$) from which each PFT can access soil water for transpiration. The simulated latent heat flux and gross primary productivity have been found to be especially sensitive to these parameters in previous studies (Blyth et al., 2010).

The full set of equations within the JULES model is documented in the papers by Best et al. (2011) and Clark et al. (2011), but the key equations are highlighted below. In JULES , leaf-level photosynthesis and stomatal conductance are treated with a coupled model (Cox et al., 1998) . Based on the models of Collatz et al. (1991, 1992), leaf-level photosynthesis $A$ is controlled

by the carboxylation rate (which depends on $n_0$, $T_{low}$, $T_{upp}$) and light-limited photosynthesis (which depends on $\alpha$). It follows that:

$$A = A(n_0, \alpha, T_{low}, T_{upp}, c_i, \beta) \tag{1}$$

where $c_i$ is the internal $CO_2$ concentration inside the leaf, and $\beta$ is a soil moisture stress factor which depends on the vertical soil moisture profile $\theta$, and the plant rootdepth $d_r$:

$$\beta = \beta(\theta, d_r) \tag{2}$$

The internal $CO_2$ concentration $c_i$ is assumed to be dependent on the external $CO_2$ concentration $c_a$ and the atmospheric humidity deficit $\delta q$ (Cox et al., 1998) via the equation:

$$\frac{c_i - c_*}{c_a - c_*} = f_0 \left( 1 - \frac{\delta q}{\delta q_c} \right) \tag{3}$$

where $c_*$ is the $CO_2$ compensation point, and $f_0$ and $\delta q_c$ are parameters that are calibrated in this study. The stomatal con-

ductance for water vapour $g$ is diagnosed in JULES from the leaf-level photosynthesis $A$ and the internal and external $CO_2$ concentrations:

$$g = 1.6 \frac{A}{c_a - c_i} \tag{4}$$

The factor of 1.6 converts the stomatal conductance for $CO_2$ into a stomatal conductance for water vapour. The scaling-up from leaf to canopy level in this version of JULES uses a "big-leaf" approach (Cox et al., 1999).

## 2.2 Data assimilation system

The term 'Data Assimilation' is commonly used to describe the process of using observations to refine the initial state within a numerical representation of a system (Bouttier and Courtier, 1999). This is most obviously the case for weather forecasting,

in which the temperature, humidity and wind fields define the initial state. However, data assimilation techniques have also been used for parameter estimation, for example in hydrological models (Madsen (2003), Liu and Gupta (2007)), and carbon cycle data assimilation systems (CCDAS; Rayner et al. (2005), Kaminski et al. (2013)). In parameter optimisation by data assimilation, the internal parameters of a model take on the role of the dynamical state variables in initial state estimation by data assimilation. Nevertheless, the underlying techniques (e.g. of defining a model adjoint and minimising the error in the fit to data), are very similar in these two applications of data assimilation. This paper is certainly not the first to define parameter estimation of this form as data assimilation (Braswell et al. (2005), Stöckli et al. (2008), Verbeeck et al. (2011), Kuppel et al. (2012), Hararuk et al. (2014)), but the reader should note the subtle difference between our definition of data assimilation and that commonly used in weather forecasting.

Even a relatively simplistic land-surface representation such as JULES has over a hundred internal parameters representing the environmental sensitivities of the various land-surface types and PFTs within the model. In general these parameters are chosen to represent measurable 'real world' quantities (e.g. aerodynamic roughness length, surface albedo, plant root-depth). This allows observationally-based estimates of these parameters to be made in the early stages of the model development process. However, the detailed performance of a land-surface model can be very sensitive to such internal parameters. It is therefore common for land-surface modellers to calibrate their models against available observations, such as eddy covariance flux data. This is typically carried out in a rather ad hoc manner with the modeller varying the parameters that he/she believes are most relevant to the model performance. Such model tuning is by its very nature subjective, lacks reproducibility, and is often sub-optimal because the modeller is unable to explore the full feasible parameter space through such a manual technique.

This paper describes a more objective approach to land-surface model calibration, adopting ideas from the applied mathematics of data assimilation as used widely in weather forecasting and other disciplines, and motivated by pioneering attempts at carbon cycle data assimilation (Rayner et al. (2005); Kaminski et al. (2013)). It utilises the adjoint of the JULES model, derived by automatic differentiation, which enables efficient and objective calibration against observations. Importantly, adJULES also allows the uncertainties in the best-fit parameters to be estimated. Such uncertainties are important information for model users, and can also form the basis for observation-constrained estimates of posterior probability density functions for the land-surface parameter perturbations used in climate model ensembles (e.g. Booth et al. (2012)).

### 2.2.1 The theory of adJULES

JULES generates a modelled time-series for a given vector of internal parameters, $z$. The cost function, $f(z)$, consists of a weighted sum of squares of the difference between $m_t$ (the vector of model outputs at time $t$), and $o_t$ (the vector of observations at time $t$), combined with a term quadratic in the difference between parameter values $z$ and initial parameter values $z_0$:

$$f(z; \hat{z}, z_0) = \frac{1}{2} \left[ \sum_t (\mathbf{m}_t(z) - \mathbf{o}_t)^T \mathbf{R}(\hat{z})^{-1} (\mathbf{m}_t(z) - \mathbf{o}_t) + \lambda (z - z_0)^T \mathbf{B}^{-1} (z - z_0) \right]. \quad (5)$$

Here, $\mathbf{R}(\hat{z}) = \frac{1}{n} \sum_{t=1}^{n} (\mathbf{m}(\hat{z})_t - \mathbf{o}_t)(\mathbf{m}(\hat{z})_t - \mathbf{o}_t)^T$ denotes the error cross product matrix produced by a JULES run with parameter value $\hat{z}$. In an optimisation, $z$ and $\hat{z}$ are updated separately in nested loops, having both been initialised to the default JULES parameter value $z_0$. In the inner loop, $z$ is varied to minimise the cost function (termination criterion: $\nabla f \approx 0$)

for the current value of $\hat{z}$. In the outer loop, $\hat{z}$ is reset to the new value of $z$ from the inner loop (termination criterion: change in $\hat{z}$ negligible). At the end of an optimisation, therefore, the matrix $\mathbf{R}$ conveys information about the error correlation structure in a JULES run with optimal parameter values.

The matrix $\mathbf{B}$ describes the prior covariances assigned to the parameters, and is here chosen to be a diagonal matrix proportional to the inverse square of the ranges allowed for each parameter. The prior uncertainties are therefore assumed to be uncorrelated between the parameters. The constant of proportionality $\lambda$ controls the relative importance of the background term (i.e. the right-hand term in Eq. 5) and the error term (i.e. the left-hand term in Eq. 5). Larger values of $\lambda$ help condition the problem and force parameter values to be close to the initial value $z_0$ (Bouttier and Courtier, 1999). All parameters and observations are equally weighted in this cost function.

The optimal vector of parameters is the vector $z$ that minimises the cost function (Eq. 5). The aim of adJULES is to find this vector. adJULES minimises the cost function iteratively using the gradient descent algorithm L-BFGS-B (Byrd et al. (1995), optim: R Development Core Team (2015)). This algorithm is based on the BFGS quasi-Newton method but is modified to use limited memory, for computational affordability, and box constraints, so each parameter is given an upper and lower bound based on expert opinion or on physical reasoning (Byrd et al., 1995).

At each iteration, the gradient $\nabla f(z)$ of the cost function $f(z)$ is computed with respect to all parameters, using the adjoint model of JULES. The adjoint is generated with the automatic differentiator tool TAF (Transformation of Algorithms in Fortran; see Giering et al. (2005)). Automatic differentiation relies on using the chain rule, the choice of forward or reverse mode refers to the order in which the derivatives are computed. Calculating $\nabla f(z)$ is most efficient in reverse mode as only one sweep is needed to generate the derivative with respect to all parameters (Bartholomew-Biggs et al., 2000).

Once the cost function reaches the minimum, a locally optimal parameter vector $z_1$ is returned and the second derivative of the cost function with respect to the parameters can be used to calculate posterior uncertainties. This process is then repeated, the locally optimised parameters are fed back through JULES, generating a new modelled time-series and hence a new cost function. The loop terminates when the modelled time series no longer improves (Fig. 1).

### 2.2.2 Multisite Implementation

In its simplest form, adJULES runs at a single grid-point location and so the derived optimal parameter vector is site-specific. On the other hand, multisite optimisation aims to find values for a common set of parameters, using data from multiple locations. The definition of the cost function (Eq. 5) can be extended to include the observations from all $S$ sites, and its derivative found in order to use the L-BFGS-B algorithm again. The extended cost function is the sum of the individual cost functions for each site $s$. Similarly, the first and second derivatives of this new cost function can be defined using the sum of the derivatives at the individual sites.

$$f(z; \hat{z}, z_0) = \frac{1}{2}\left[\sum_s \sum_t (\mathbf{m}_{t,s}(z) - \mathbf{o}_{t,s})^T \mathbf{R}_{\mathbf{s}}(\hat{z})^{-1}(\mathbf{m}_{t,s}(z) - \mathbf{o}_{t,s}) + S\lambda(z - z_0)^T\mathbf{B}^{-1}(z - z_0)\right]. \tag{6}$$

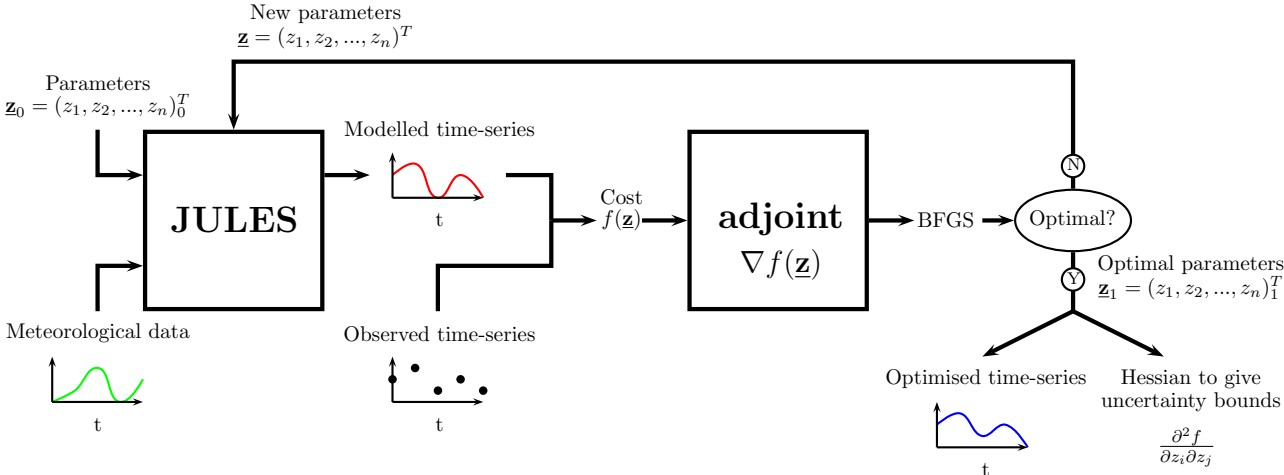

**Figure 1.** Schematic of the adJULES parameter estimation system starting with the initial parameter vector $z_0$. This is usually based on default JULES parameter values (Blyth et al., 2010). The optimised parameter vector is denoted $z_1$.

An additive cost function, where the optimisation criterion is to minimise the total cost, was chosen over a cost function where all individual cost functions are required to be improve. All of the sites were used in finding the optimal parameter vector for each PFT, so that sites which do not improve with the rest of the PFT suggest incorrect classification of the site or issues with the PFT definitions.

## 2.3  Eddy covariance flux data

The eddy-covariance flux data used in this study are part of FLUXNET (Baldocchi et al., 2001). The FLUXNET database contains more than 500 locations worldwide, and all of the data are processed in a harmonised manner using the standard methodologies including correction, gap-filling and partitioning (Papale et al., 2006). Data from 160 sites were made available for this study by M. Groenendijk. The sites used in this study were selected based on data availability: sites with missing input variables or data gaps of more than 50% during the growing season were omitted.

To constrain photosynthetic parameters, Net Ecosystem Exchange (NEE) and Latent Heat Flux (LE), among other fluxes, are helpful. The NEE flux, defined as the net flux of $CO_2$, is partitioned into gross primary production (GPP) and ecosystem respiration (Resp) (Reichstein et al., 2005). In this study the GPP flux is used along with the LE flux to constrain the model. GPP data are model-derived estimates, which could introduce an additional uncertainty into the results. Also note that due to model structural errors, calibration against these two particular observables could cause model simulations of other fluxes (not used in the tuning) to become worse Gupta et al. (1999).

In an attempt to run the experiments as closely to a standard JULES run as possible, input fields of vegetation structure and soil type were drawn from the UK Met Office ancillary files used in the HadGEM2 configurations. The LAI seasonal cycle used

is derived from a MODIS product (Myneni et al., 2002) from Boston University. The values taken for each of the experiment sites correspond to the closest grid point at which data are available. This could lead to inconsistencies between the actual vegetation at a given site and the vegetation structure and soil type used in the model.

## 2.4 Experimental setup

Version 2.2 of JULES is implemented in the current version of adJULES. This version is set up to calibrate a subset of JULES soil and vegetation parameters against up to six observables in the vectors $m_t$ and $o_t$ (Eq. 5): net ecosystem exchange (NEE), sensible heat (H), latent heat (LE), surface temperature ($T_s$), gross primary productivity (GPP) and ecosystem respiration (Resp).

This study aims to improve the parameters used to define PFTs and therefore it concentrates on vegetation parameters. Table 1 outlines the parameters chosen. Note that this only represents a modest subset of the parameters available and as such the results could be different when considering different subsets in the calibration.

One year of FLUXNET data is used for each site considered in this study at the calibration stage. Where multiple years are available, the most complete year was chosen. For each site the model is spun up to a steady soil moisture and temperature state. Where possible, the two years of data preceding the year of comparison were applied repeatedly in the spin-up. Where this was not possible, the first year of data was repeatedly applied. Only sites with at least two years of data are used in this study, so that the spin-up year is different from the experiment year. In each case, the model was spun up for at least 50 years. For deciduous sites and crop sites, leaf area index values are taken from MODIS data for the appropriate year. Where possible, a second year of FLUXNET data was spun up to be used at the validation stage of this study. This second year was chosen to be the second most complete year when more than one year was available.

The sites used in each of the PFT classes are described in Appendix A. The FLUXNET database used in this study did not distinguish between the different types of grasslands. Using Met Office ancillary files, the grasslands were partitioned into C3 grasses and C4 grasses according to fractional cover. In the case of C3 grasses, sites were picked only when the fractional cover was over 60%. Since the C4 grasses are under-represented in the FLUXNET database, this boundary was lowered to include all sites where C4 grass was the dominant PFT. Crops were not included in either grass class. The photosynthesis model used in JULES is based on scaling up observed processes at the leaf scale to represent the canopy. The scaling to canopy level can be done in several ways. In this study the simple big leaf approach was adopted (Clark et al., 2011), although optimisations can also be carried out for more complex canopy radiation options (Mercado et al., 2009).

All of the sites in each PFT class are used to find the optimal values for the PFT. The second derivative of the cost function found by differentiation of the adjoint code is then used to quantify the uncertainties associated with these optimal parameter vectors.

Preliminary experiments showed very narrow uncertainties whilst running the optimisation scheme over multiple sites (i.e. the background term was found to dominate the cost function). In previous multisite studies (Kuppel et al., 2012, 2014), the prior range was also used to defined the background covariance matrix $\mathbf{B}$. The range was variously further multiplied by a factor of 40% (Kuppel et al., 2012) and $\frac{1}{6}$ (Kuppel et al., 2014). Experiments were run to find a similar factor to use in this

study (the constant of proportionality $\lambda$ in Eq. 5). In each of the multisite experiments, the lowest value of $\lambda$ such that the Hessian is positive definite at the optimal parameter value was used. This allows uncertainties to be generated around each parameter and prevents the gradient descent algorithm from reaching the boundaries of the prescribed prior range.

## 2.5 Analysis tools

### 2.5.1 Parameter uncertainty

As well as generating optimal parameter values, adJULES estimates the uncertainty associated with each parameter. The second derivative (Hessian) of the cost function,

$$H_{ij} = \frac{\partial^2 f}{\partial z_i \partial z_j} \tag{7}$$

where $f(z)$ is given by equation (5), evaluated at the optimal parameter value, yields information about the curvature of the cost function at the local minimum. A 'sharp' cost function, where the cost function is steep either side of the optimal parameter value, indicates lower parameter uncertainty. This can also be interpreted as meaning that a small deviation from the optimal parameter value yields a large increase in cost. Conversely, a 'flat' cost function indicates higher parameter uncertainty, or little change in cost caused by deviation from the optimal parameter value.

In order to generate statistics associated with the curvature of the cost function, the Hessian is used to generate samples from the posterior distribution. This is a truncated multivariate normal distribution (Genz et al., 2015) because of the box constraints placed on the prior. Using Gibbs sampling (Geman and Geman, 1984), an ensemble of plausible parameter vectors is generated from this distribution, for a statistically satisfactory match between observations and modelled time series. The multivariate normal parameter distribution allows for marginal density plots to be generated for each parameter. When considering these marginal density plots, it is important to remember that they represent only one or two dimensions of a high dimensional multivariate normal distribution which is truncated. Consequently the optimal parameter values (which are modes of the full high dimensional distribution) may not coincide with modes of the one- and two-dimensional marginal distributions.

In order to illustrate the parameter uncertainties, error bars are used to represent the 80% quantile range (10th to 90th percentile) for each optimal parameter.

### 2.5.2 Fractional error - a metric of model-data fit

To measure the improvement exhibited by different parameter vectors, the fraction of variance unexplained $\epsilon^2$ is used to define the fractional error $\epsilon$. This metric was chosen to show not only the improvement made by the optimal parameter vectors at each site but also to show how each site performed relative to others.

Given a parameter vector, $z$, a modelled time series $m_{i,t}$ with $k$ data points is generated using JULES, where $i$ denotes one of the observable data streams (in this case LE and GPP). For each data stream $i$, the fraction of variance unexplained by the

model is

$$\epsilon_i^2 = \frac{\sum_{t=1}^{k}(\boldsymbol{o}_{i,t} - \boldsymbol{m}_{i,t})^2}{\sum_{t=1}^{k}(\boldsymbol{o}_{i,t} - \bar{\boldsymbol{o}}_i)^2}, \quad \text{where} \quad \bar{\boldsymbol{o}}_i = \frac{1}{k}\sum_{t=1}^{k}\boldsymbol{o}_{i,t} \tag{8}$$

It follows that the mean fraction of variance unexplained across data streams,

$$\epsilon^2 = \frac{\epsilon_1^2 + \epsilon_2^2}{2}, \tag{9}$$

is a single dimensionless measure of model misfit. The fractional error $\epsilon$ can than be interpreted as the typical (root-mean-square) error expressed as a fraction of the (root-mean-square) magnitude of the observed seasonal cycle. Thus, $\epsilon = 0$ represents a perfect match to the observations, while $\epsilon = 1$ corresponds to the error in a null model whose prediction $\boldsymbol{m}_{i,t}$ always equals

the observational mean $\bar{\boldsymbol{o}}_i$.

In Hydrology this is related to a metric known as the "Nash-Sutcliffe" efficiency (Nash and Sutcliffe, 1970), equivalent to $1 - \epsilon^2$, and has been used by many studies to perform cross-site comparisons.

## 3   Results and discussion

In this section, the site-specific optimisations are considered first. By considering each PFT separately, the misfits between the

model and the observations are discussed and the effect of optimising over each site individually to improve model-observation agreement is considered.

Next, the multisite methodology is used to perform optimisations over each of the PFTs. All of the sites in a given PFT are optimised simultaneously to find a generic parameter vector appropriate to the PFT. The new optimised parameter vectors are presented, along with associated uncertainties. Some of the uncertainties and correlations found between parameters are

discussed, especially in the context of the equations described in section 2.1. The rest of the section considers the improvement found using these optimised parameter vectors both on the calibration year and the validation year for each of the sites.

### 3.1   Single-site optimisations

First, each of the sites was optimised individually in order to find site-specific parameter vectors. Typically, this required about 150 function evaluations to find a local optimum. As described in section 2.4, one year runs at the different sites were

optimised against monthly averaged LE and GPP. The constant of proportionality $\lambda$ is set to 1 for all sites, in order to give equal weighting to both terms in Eq. 5. A site dominated by each PFT was picked to represent the general improvements made. The main seasonal cycles of LE and GPP for the different sites are shown in Fig. 2.

Most broadleaf sites follow the pattern illustrated (Fig. 2, top row). Normally, for broadleaf sites, a standard JULES run will underestimate GPP. The optimisation does a good job in correcting this, bringing the modelled time-series closer to the observations. In contrast, LE does not improve as much.

Similarly for the needleleaf sites (Fig. 2, second row), the JULES model output tends to overestimate LE and underestimate

GPP. The parameter vector found in the optimisation improves the fit of both data streams, most notably GPP. At sites for which a double peak seasonality is apparent, the optimised model captures this better than the original model.

GPP is also underestimated for the C3 grass sites (Fig. 2, middle row) and, for the majority of the sites, the optimisation does a good job of correcting this. The LE flux tends to have the right magnitude before optimisation, unlike the GPP flux, but adJULES does not manage to improve this output significantly. In the example shown, the JULES model using the default

parameter vector already performs very well, so little improvement is needed, but this is not always the case. The new set of parameters is also good at simulating multiple peaks in the LE and GPP fluxes, when they are observed.

There are only two C4 grass sites in the set and JULES does not perform very well on these before or after optimisation (Fig. 2, fourth row). The original stomatal conductance-photosynthesis model within JULES was developed based on fluxes measured over C4 grass as part of the FIFE field experiment (Cox et al., 1998). However, there are relatively few FLUXNET

sites over C4-dominated landscapes, and only two even in the extended dataset used here. As a result, the sensitivity of stomatal conductance and photosynthesis to environmental factors has been less well tested for C4 grasses. The results presented in this paper therefore highlight the need to reassess JULES and other land-surface models for predominantly C4 landscapes.

The shrub sites show no general pattern (Fig. 2, fourth row). Some sites overestimate LE, whilst others underestimate it, and similarly for GPP. The level of improvement varies over sites. For some of the sites in this PFT, the magnitude of GPP fails

to get close to the magnitude of the observations, both before and after optimisation. However, it is hard to pick out a general pattern for this PFT, since there are only 5 sites in this set.

Overall, the adJULES system works well in finding optimal parameter vectors which improve the performance of JULES at individual sites, regardless of PFT. The systematic underestimation of GPP in default JULES improves the most. This larger improvement in GPP fit reflects the larger set of optimised parameters that are exclusively related to the carbon cycle. Different

parameters may need to be incorporated, for example some soil ones, for the LE flux to improve further.

## 3.2 PFT-specific optimal parameter values

Optimisations were performed over all available sites for each of the PFTs simultaneously. The optimised model parameters for each of the PFTs are presented in Fig. 3.

For half of the parameters, the prior parameter value lies outside the posterior uncertainty bounds. The $\frac{\delta c}{\delta l}$ parameter, which

determines the efficiency of rainfall interception by the plant canopy, does not change much from its original value for any of the PFTs. The uncertainty bounds are relatively tight and symmetrical. The rest of the parameters show more variation. As described in section 2.5.1, the optimal values need not be in the centre of the uncertainty range, the PDF can be skewed. Most of the PFTs display high uncertainty in at least one of the parameters optimised; for the optimised broadleaf set for example, $dq_c$ is highly unconstrained. For C4 grasses, $d_r$ is so unconstrained that the optimal value found lies outside the 80% confidence interval. C3 grasses show large uncertainty in $n_0$ and for shrubs, the parameter with the largest uncertainty is $\alpha$.

The uncertainties shown in Fig. 3 are one-dimensional marginal distributions. To understand further how the parameters are correlated, consider the two-dimensional representation in Fig. 4. For all of the PFTs, the posterior parameter uncertainties exclude a large part of the prior ranges. The cloud of plausible points tends to be restrictive and tight for most parameters.

Fig. 4 shows clear correlation of some parameters, especially for the tree PFTs. Many of these correlations can be understood in terms of the underlying structure of the JULES model (Section 2.1). For example, the positive correlation of $n_0$ with $f_0$, and the negative correlation of $n_0$ with $dq_c$, are consistent with adJULES attempting to fit the stomatal conductance $g$, which controls the transpiration flux from taller vegetation. The stomatal conductance has the approximate form

$$g \approx 1.6 \frac{A}{c_a} \left( \frac{1}{(1 - f_0) + f_0 \frac{dq}{dq_c}} \right) \tag{10}$$

if it is assumed that $c_* \ll c_i$ and $c_* \ll c_a$.

The maximum rate of leaf photosynthesis is controlled largely by the leaf nitrogen content $n_0$, especially in this big-leaf version of JULES (Cox et al., 1999). The best fit parameters for tree PFTs also seem to imply that the second term in the denominator dominates over the first. As a result, maintaining a realistic $g$ value, and therefore a realistic latent heat flux, will require that $n_0$ and $f_0$ vary proportionally, and that $n_0$ and $dq_c$ values are negatively correlated. This is consistent with Fig. 4(a) and (b).

Such correlation of parameters is less obvious for the grass PFTs, because evapotranspiration is controlled less by stomatal conductance and more by the smaller aerodynamic conductances associated with shorter vegetation.

Choice of $\lambda$ had less effect on the values of the optimal parameters than on uncertainties and correlations found. The uncertainty ranges become larger with smaller values $\lambda$ and correlations less pronounced.

### 3.3 Assessment of PFT-specific optimal parameters

The performance of the PFT-specific parameters is now compared to the default JULES values and to the parameters found by optimising independently at each measurement site. For each site, the fractional error in both the calibration year and the validation year is displayed Fig. 5.

By definition, the fractional error in calibration years decreases when moving from default to site-specific optimal parameters in the calibration years. Remarkably, the site-specific optimal parameters also improve the model-data fit in validation years for 59/64 (92%) of sites. Similarly, the PFT-specific optimal parameter vector improves the fit (in both calibration and validation years) for 85% of the sites; 75/79 sites for the calibration years and 55/64 sites for the validation years.

Consider first the broadleaf sites (Fig. 5, top two rows). For the majority of sites displayed in the top broadleaf panel, the reduction in fractional error in moving from default to site-specific optimal parameters is substantial and sometimes as much as a factor of 2. In the calibration year, the PFT-specific optimal parameter vector improves 26 of the 27 broadleaf sites shown although one of the sites, IT-Lec, the fit shows no change. The improvement is typically about half as good (on a log scale) as the improvement using the site-specific optimal parameters. In other words, the reduction in fractional error moving from default to PFT-specific optimal parameters is sometimes as much as a factor of $\sqrt{2}$. Amongst broadleaf sites, only UK-PL3 gets notably worse. Investigation shows that this site behaves differently from the rest of the sites in the set, both in the magnitude of

the fluxes and seasonality. This UK site is in the Pang/Lambourn catchment, which has chalk soil with macropores that permit significant lateral subsurface flows of soil moisture. These horizontal flows cannot be captured in a model like JULES which is essentially one-dimensional in the vertical below the soil surface.

Similar levels of fit and error reduction can be seen in the validation years in the Broadleaf set. Only IT-Col and US-MMS show no improvement, the PFT-specific optimal parameter vector does not worsen the fit at these locations. For AU-Tum, the PFT-specific parameter vector outperforms the site-specific vector. This illustrates that the PFT-specific vector can be robust, whereas the locally-optimised vectors might over-tune to the specific behaviour of the calibration year.

Results are similar for the Needleleaf sites, the majority of the sites show noticeable improvements in both the calibration and

validation years when using site-specific optimal parameter vectors. For over a third of the sites in this PFT, the improvement when using the PFT-specific parameter vector is similar to that obtained with the site-specific parameter vector. This illustrates that these sites fit well together as a single PFT. For these sites, the PFT-specific vector sometimes outperforms the site-specific vector on the validation years. Some sites in the Needleleaf PFT remain unchanged regardless of the parameter vector used. Anomalous sites that should be noted are CA-Qcu, CA-SF3 and US-Blo. The CA-Qcu site is the only one in this PFT that

does not improve when using the PFT-specific vector, for either the calibration or validation years. This site has a lower annual cycle of GPP than the rest in this set. The CA-SF3 site improves when using the site-specific parameter vector in the validation year, but not using the PFT-specific vector. The US-Blo site improves in the calibration year, but when confronted with the validation year, both the site-specific vector and PFT-specific vector worsen the fit. This validation year has unusually high LE which might be causing this discrepancy.

For some sites, (e.g. US-Blo and BW-Ma1), the PFT-specific optimum outperforms the site-specific optimum in the calibration year. This phenomenon was also noted by Kuppel et al. (2014), who suggest that the added constraints placed on the parameters by increasing the number of sites causes the cost function to become 'smoother'. This may render the optimisation scheme less likely to become trapped in local minima.

The last panel of Fig. 5 shows the C3 grass sites, the C4 grass sites and the Shrub sites. For the C3 grass sites, the majority

of the validation years have a better fit with the PFT-specific parameter vector than with site-specific parameter vector. This suggests that the seasonal cycle differs over the different years at these sites. For the C4 grass sites, which started with relatively high errors, the new parameter vectors improve the sites slightly for the calibration year but hardly at all for the validation year. This set of two sites is too small to draw any proper conclusion about the C4 grass parameters. There is a clear need for more data from C4 grass sites. Finally, the Shrubs can be seen to improve for all the sites. For the Shrub sites, both the site-specific

and the PFT-specific provide a better fit of the model to the observations of the calibration year. The improvement is minor for these sites, except for CA-Mer which halves its fractional error. When confronted with observations from the validation years, the model also improves the fit of these sites for both site-specific and PFT-specific parameters (with the exception of US-Los, where the site-specific optimal vector increases error but the PFT-specific vector reduces it). This is another example of the PFT-specific parameter vector being more robust.

### 3.4 Analysis of improvement in fit

The fractional error is a good tool for cross-site comparison but it does not give much information about the way in which the optimised parameter vectors improve the fit at each site. Taylor diagrams (Taylor (2001)) provide more insight into how the fit has been improved by considering the relationship between observed variance $var(\mathbf{o}_t)$, modelled variance $var(\mathbf{m}_t)$, error variance $var(\mathbf{o}_t - \mathbf{m}_t)$ and model–observation correlation $cor(\mathbf{o}_t, \mathbf{m}_t)$.

The Taylor diagrams in Fig. 6 illustrate the improvement in performance of the optimised model for both the site–specific and PFT–generic parameters during calibration years (plots for validation years are very similar).

For latent heat at broadleaf sites (left), the improvement is most noticeable in cases where the seasonal cycle was over-estimated. The correlation between the modelled time-series and observation time-series does not improve much but for the majority of the sites this starts off relatively high (over 0.6). Other PFTs show less drastic improvements for latent heat.

For GPP at needeleaf sites (right), the seasonal cycle is typically underestimated and improves notieceably for both the single–site parameter vectors and the PFT–generic parameter vectors. The correlation between model and observed time-series does not change greatly. The Taylor diagram for GPP at broadleaf sites is very similar. For grasses and shrubs, the change is less drastic, though some of the sites have a more notable increase in correlation.

Since Taylor diagrams are based on a decomposition of the variance of the errors they are insensitive to any systematic offset in the model. It therefore makes sense to consider in addition the normalised bias $|\mu_m - \mu_o|/\sigma_o$. Calculating this statistic separately shows a reduction in bias for nearly all sites. Taken together, these measures show that the observed improvements in model fit are due mainly to adjustment of the magnitude of the annual cycle and a reduction in bias.

## 4 Conclusions

This study introduces the adJULES system, which has been developed to tune the internal parameters of the JULES land surface model. adJULES enables objective calibration of JULES against observational data, providing best fit internal parameters and associated uncertainty ranges.

For individual FLUXNET sites, adJULES has the ability to find local (site-specific) optimal parameter vectors that significantly improve the performance of the JULES model compared to runs generated using the default parameters. The data streams used in the calibration, LE and GPP, are both modelled more accurately with the optimal parameter vectors, with the GPP flux improving the most. The greater improvement in the GPP flux is due to the fact that the parameters considered in this study are mainly related to photosynthesis. For the LE flux to improve more significantly, more water and energy related parameters would need to be considered in the optimisation.

When optimised locally to find site-specific parameters, all of the sites in this study improve the model-data fit for the calibration year. In addition, when confronted with independent data from a validation year, the locally optimised parameter vectors decreased the error in model-data fit for 92% of the sites. This validation of the site-specific parameter vectors is promising, and suggests that the adJULES system is robust. It also gives confidence that the parameter vectors found can be generalised over different locations.

This study is motivated partly by the desire to improve the performance of JULES within the Hadley Centre's Earth System Models, which means needing to find best fit parameters for a relatively small number of PFTs. The adJULES system has the ability to calibrate multiple locations simultaneously in order to find best-fit parameters. This 'multisite' optimisation is a relatively new feature in terrestrial data assimilation. By classifying the FLUXNET sites into groups dominated by each JULES PFT (BT, NT, C3G, C4G, Sh), adJULES was used to find the optimal PFT-specific parameters.

Although the PFT-specific optimal parameters do not always fit the data as well as site-specific optimal parameters, they still offer significant improvements over the default JULES parameters. For over 85% of the sites, PFT-specific optimal parameters perform better than default parameters when confronted with independent validation data. For 50% of the sites, the PFT-specific optimal parameters perform at least as well as site-specific optimal parameters. This implies that the multisite methodology is less susceptible to over-tuning, both in terms of variablity across sites (e.g. different overground biomass and tree ranges), and in terms of variability through time (e.g. unusually high rainfall in the calibration year).

The PFT-specific parameters found in this study represent a significant improvement on the default ones. The fact that such parameters could be found implies robust parameterisations independent of geography. This supports the idea that it is possible to represent global vegetation with a relatively small number of PFTs.

A successful and robust multisite optimisation assumes that sites can be grouped and parameter values can apply to several sites at once. Whilst the PFT-specific parameters show great improvement, agreeing with the use of five PFTs in JULES, it would be possible to rethink the PFT definitions and group sites differently. This could be done either by looking more closely at the site specifics detailed in the FLUXNET database, or by considering single-site optimisations and performing a cluster analysis in parameter space to identify PFTs empirically.

There is always a risk of becoming stuck in local minima when optimising within a high-dimensional parameter space by gradient descent. When an optimisation finds a local minimum, the final optimised state depends on the initial conditions. The consistency between our single site and multiple site optimisations therefore gives us some confidence in the robustness of the convergence of our algorithm for this application.

It is however clear that there are some limitations to the success of the optimisation results. Some sites still show significant differences between model output and observations. This suggests that improvement to model physics may be necessary in order to produce better model output. This is because adJULES produces the (locally) best possible fit to observations, given the existing model physics and the prescribed driving data. If the fit is still inadequate, this may be due to the model and data themselves, rather than parameter values. adJULES can therefore be used in the identification of model structural errors. Another reason for inadequate fit may be due to the method used. A limitation of gradient descent methods, such as the optimisation scheme used in this study, is that the local minimum found depends on the initial parameter vector. However, as discussed in section 3.3, the fact that the cost function becomes smoother with additional sites may help with becoming trapped in local minima (Kuppel et al., 2014). Alternative methods, including ensemble methods, could avoid this issue, but are more computationally costly. For some PFTs (notably C4G and Shrubs) there are insufficient FLUXNET sites to determine optimal parameters satisfactorily. Additional data and sites for these PFTs are therefore urgently required.

**Code availability**

The source code of the adJULES data assimilation system is available at http://adjules.ex.ac.uk/. The JULES land surface model is freely available to any researcher for non-commercial use. Version 2.2 used in this study can be requested at jules.jchmr.org. The main documentation for the JULES system can also be found at this site. The adjoint of the JULES model has been generated using commercial software TAF (sect. 2.2.1). For licensing reasons, the recalculation of the adjoint following code changes can be done only by the authors at the University of Exeter.

**Appendix A**

**Appendix B**

*Acknowledgements.* This work was supported by the UK Natural Environment Research Council (NERC) through the National Centre for Earth Observation (NCEO).

This study used eddy-covariance data acquired by the FLUXNET community and in particular by the following networks: AmeriFlux (U.S. Department of Energy, Biological and Environmental Research, Terrestrial Carbon Program (DE-FG02-04ER63917 and DE-FG02-04ER63911)), AfriFlux, AsiaFlux, CarboAfrica, CarboEuropeIP, CarboItaly, CarboMont, ChinaFlux, Fluxnet-Canada (supported by CFCAS, NSERC, BIOCAP, Environment Canada, and NRCan), GreenGrass, KoFlux, LBA, NECC, OzFlux, TCOS-Siberia, USCCC. Support for eddy covariance data harmonisation was provided by CarboEuropeIP, FAO-GTOS-TCO, iLEAPS, Max Planck Institute for Biogeochemistry, National Science Foundation, University of Tuscia, Université Laval and Environment Canada and US Department of Energy and the database development and technical support from Berkeley Water Center, Lawrence Berkeley National Laboratory, Microsoft Research eScience, Oak Ridge National Laboratory, University of California - Berkeley, University of Virginia.

The authors are grateful to T. Kaminski and R. Giering from FastOpt for their contribution to the development of the adjoint model, and to M. Groenendijk, A. Harper, and the UK Met Office for processing and sharing their data.

The authors are particularly grateful to two anonymous referees for their thoughtful and constructive reviews which greatly improved this manuscript.

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

**Table 1.** Parameters in optimisation vector, with descriptions.

| Symbol | Name in code | Description | Units |
|---|---|---|---|
| $n_0$ | nl0 | Top leaf nitrogen concentration | kg N (kg C)$^{-1}$ |
| $f_0$ | f0 | Maximum ratio of internal to external $CO_2$ | - |
| $d_r$ | rootd_ft | Root depth | m |
| $\alpha$ | alpha | Quantum efficiency | mol $CO_2$ per mol PAR photons |
| $\frac{\delta c}{\delta l}$ | dcatch_dlai | Rate of change of canopy interception capacity with LAI | kg m$^{-2}$ |
| $T_{\text{low}}$ | tlow | Lower temperature for photosynthesis | °C |
| $T_{\text{upp}}$ | tupp | Upper temperature for photosynthesis | °C |
| $dq_c$ | dqcrit | Humidity deficit at which stomata close | kg kg$^{-1}$ |

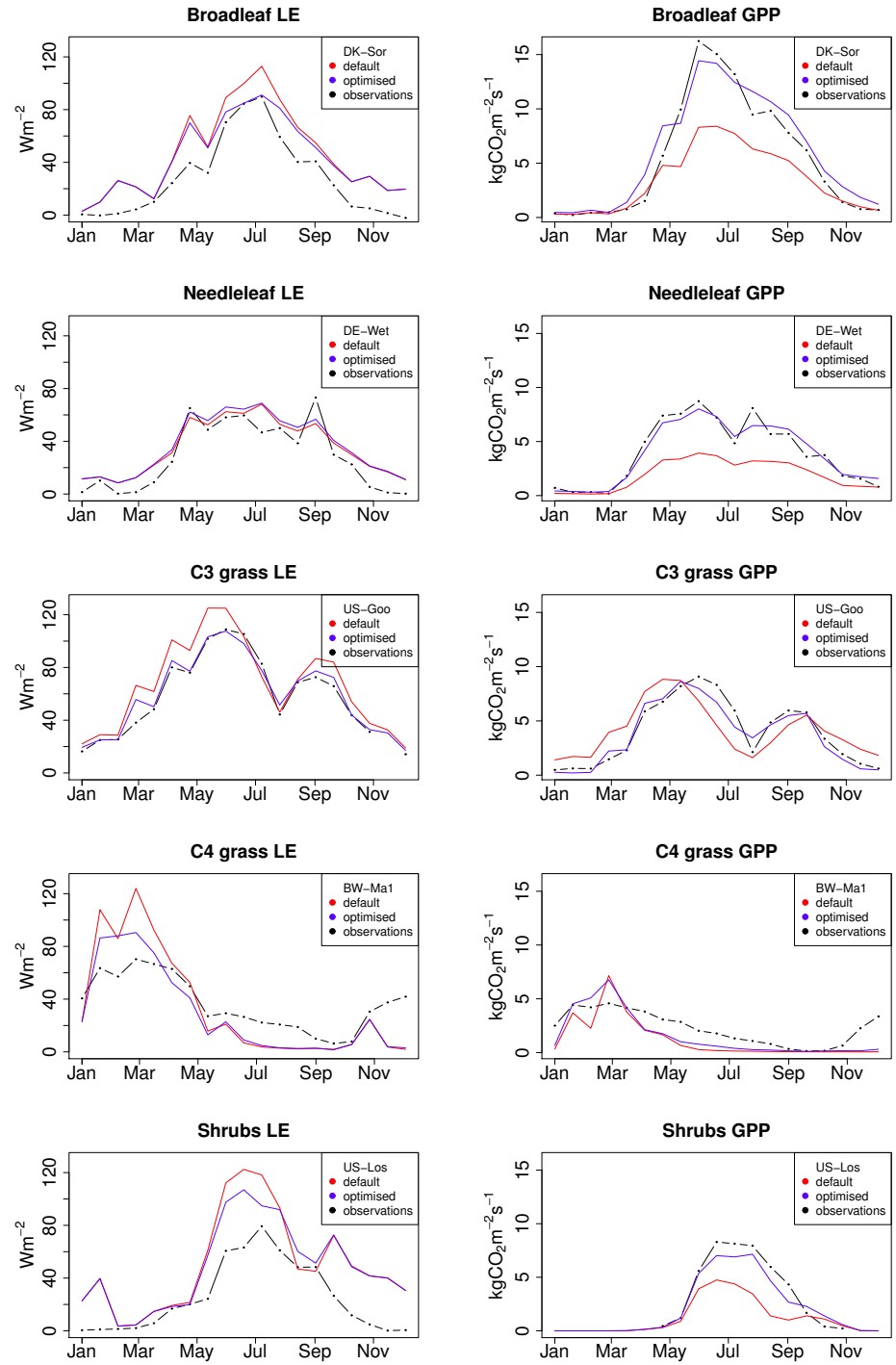

**Figure 2.** Time-series plots for illustrative site-specific validations showing LE (left) and GPP (right) for each of the different PFTs. Observations (black) are compared to JULES runs using default parameters (red) and site-specific optimal paremeters (blue).

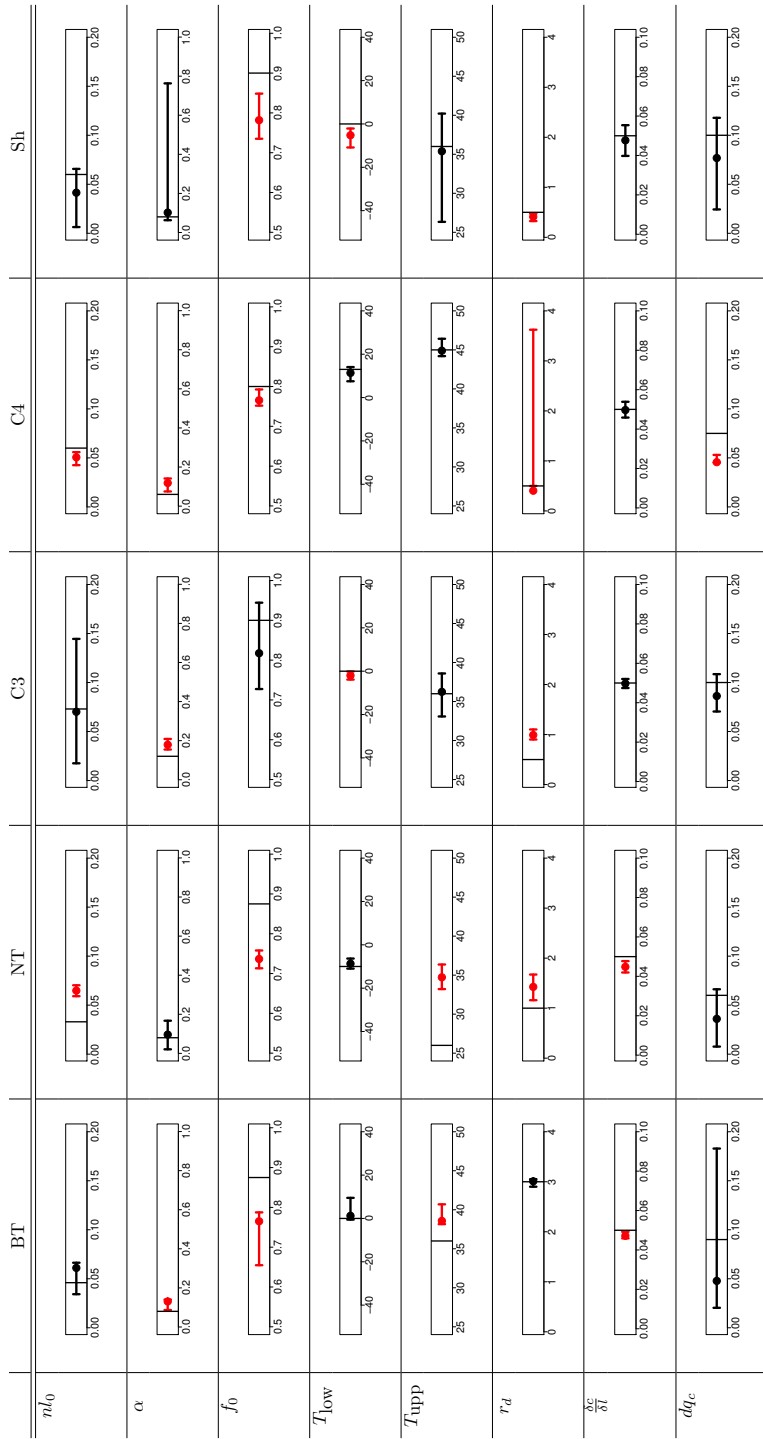

**Figure 3.** Summary of PFT-specific optimal JULES parameters found in this study (Table 1). The error bars show the uncertainty ranges given as an 80% quantile interval. The range of each box is the prior range of the parameters. Highlighted in red are the error bars for which the prior values (vertical line) are found outside the posterior uncertainty bounds. A numerical version of this figure is given in Table B1.

**Figure 4.** The correlations between parameters for PFT-specific parameter optimisations. Each subfigure shows a two-dimensional correlation map, within which each box is a 2-D marginal plot. Bar graphs show 1-D marginal distributions for individual parameters. The dimensions of the boxes represent the prior range of each parameter. Red points/dashed lines represent initial parameter values. Blue points/dashed lines represent optimised parameter values. Blue contours illustrate the posterior distribution.

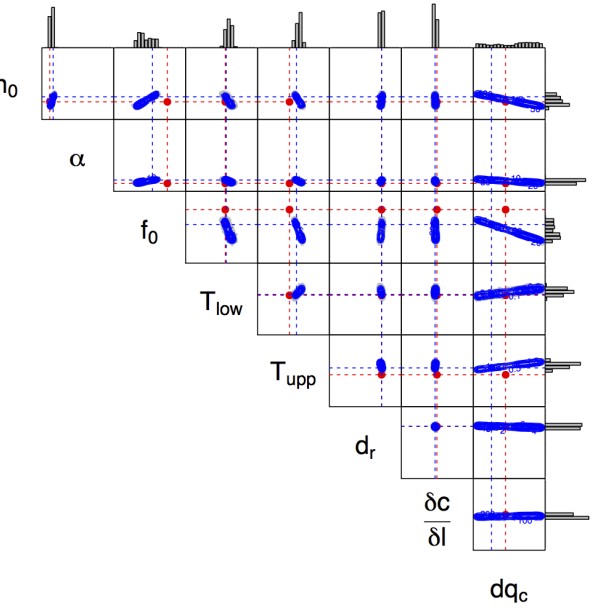

(a) Broadleaf

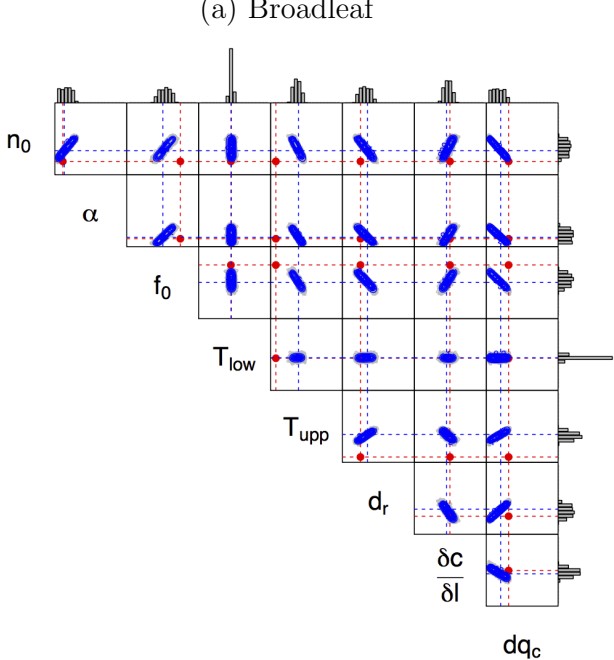

(b) Needleleaf

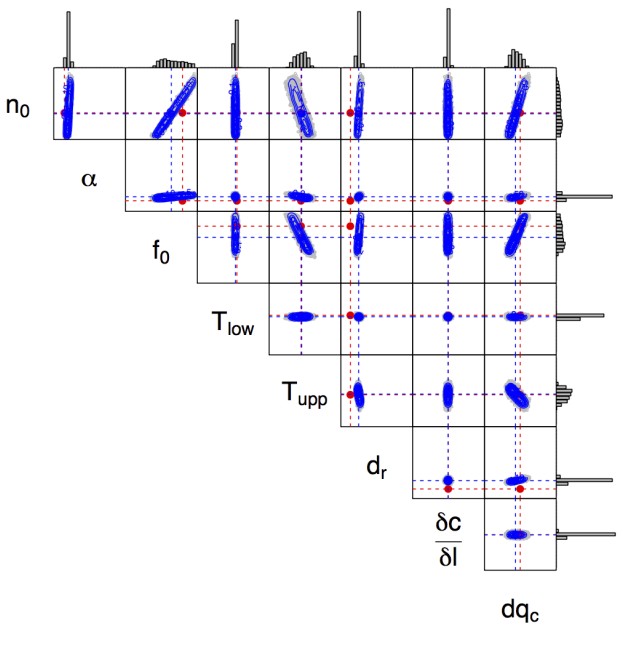

(c) C3 grasses

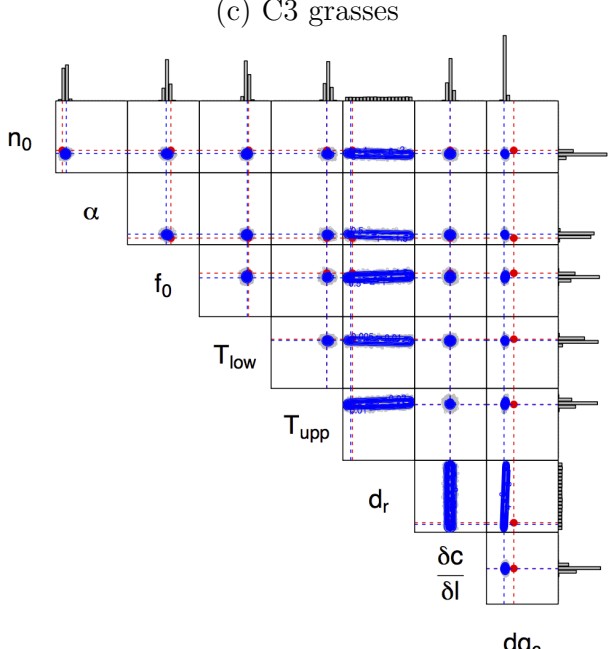

(d) C4 grasses

**Figure 4.** continued

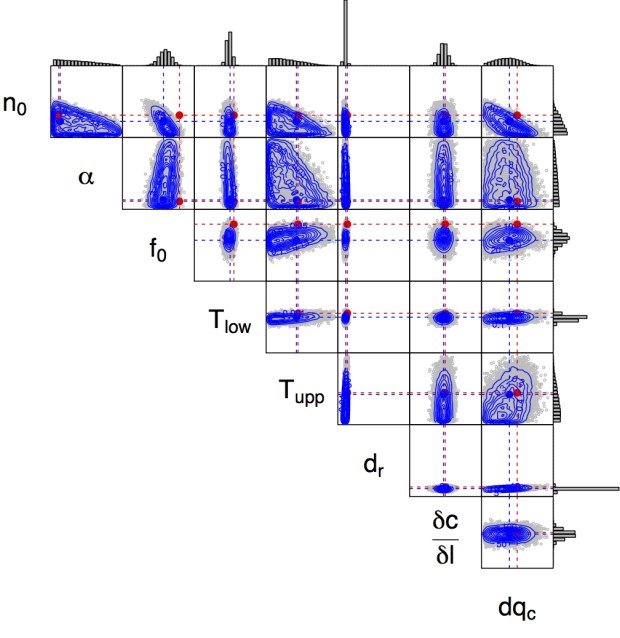

(e) Shrubs

**Figure 4.** continued

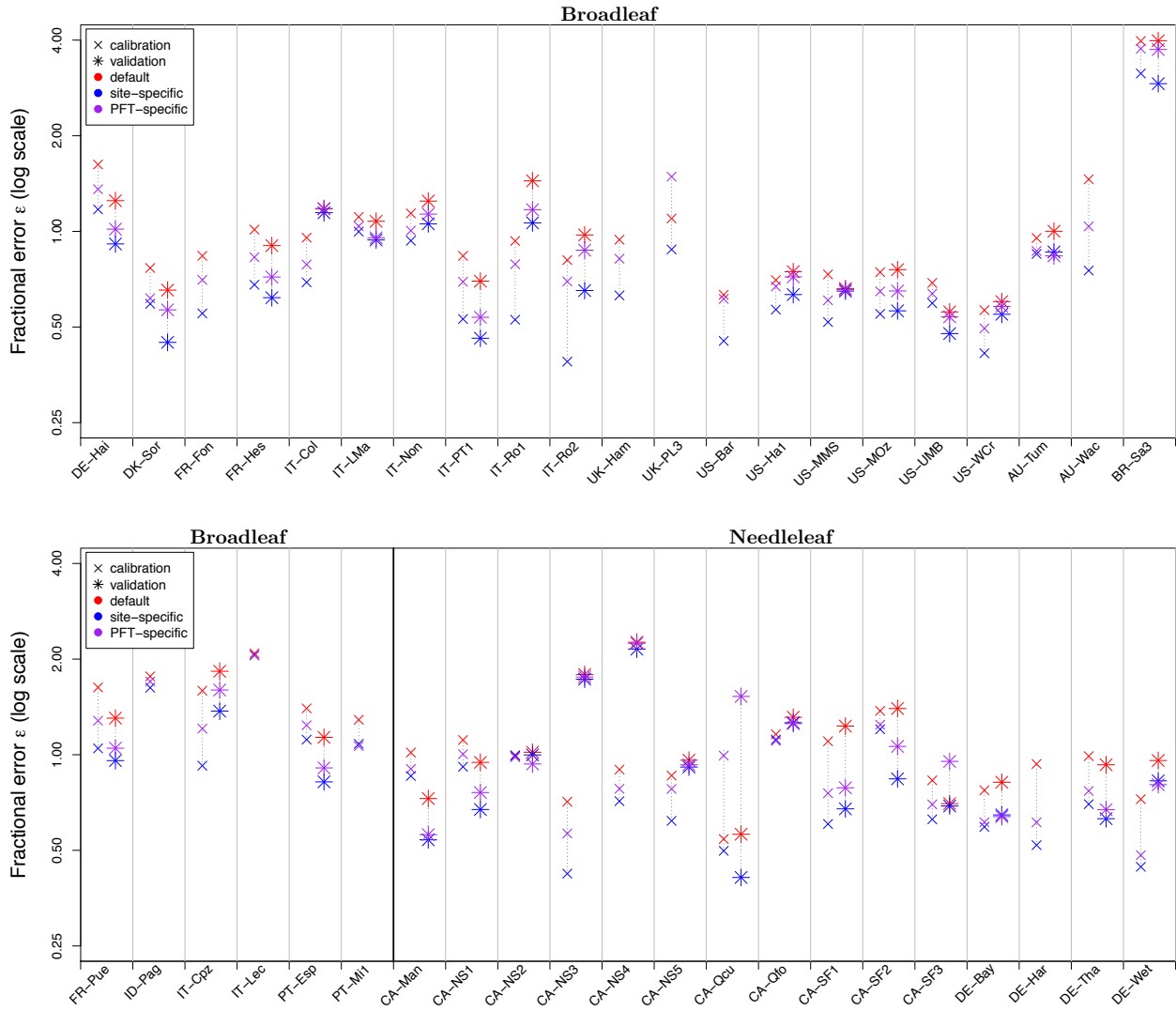

**Figure 5.** Calibration and validation of site-specific and PFT-specific parameter optimisation at FLUXNET sites, using the metric described in section 2.5.2. Fractional error shown for: default JULES parameters (∗), site-specific optimal parameters (∗), PFT-specific optimal parameters (●). Results are shown both for the calibration year (×, on left) and for the validation year (∗, on right). No validation year was available for some sites (Broadleaf: FR-Fon, UK-Ham, UK-PL3, US-Bar, ID-Pag, IT-Lec, PT-Mi1, Needleleaf: SE-Sk2, UK-Gri, US-Me4, US-SP1, Shrubs: DE-Gri, DK-Lva, PL-wet). Sites with very large initial errors have been removed from the plot (Broadleaf: BR-Sa1, Shrubs: IT-Pia).

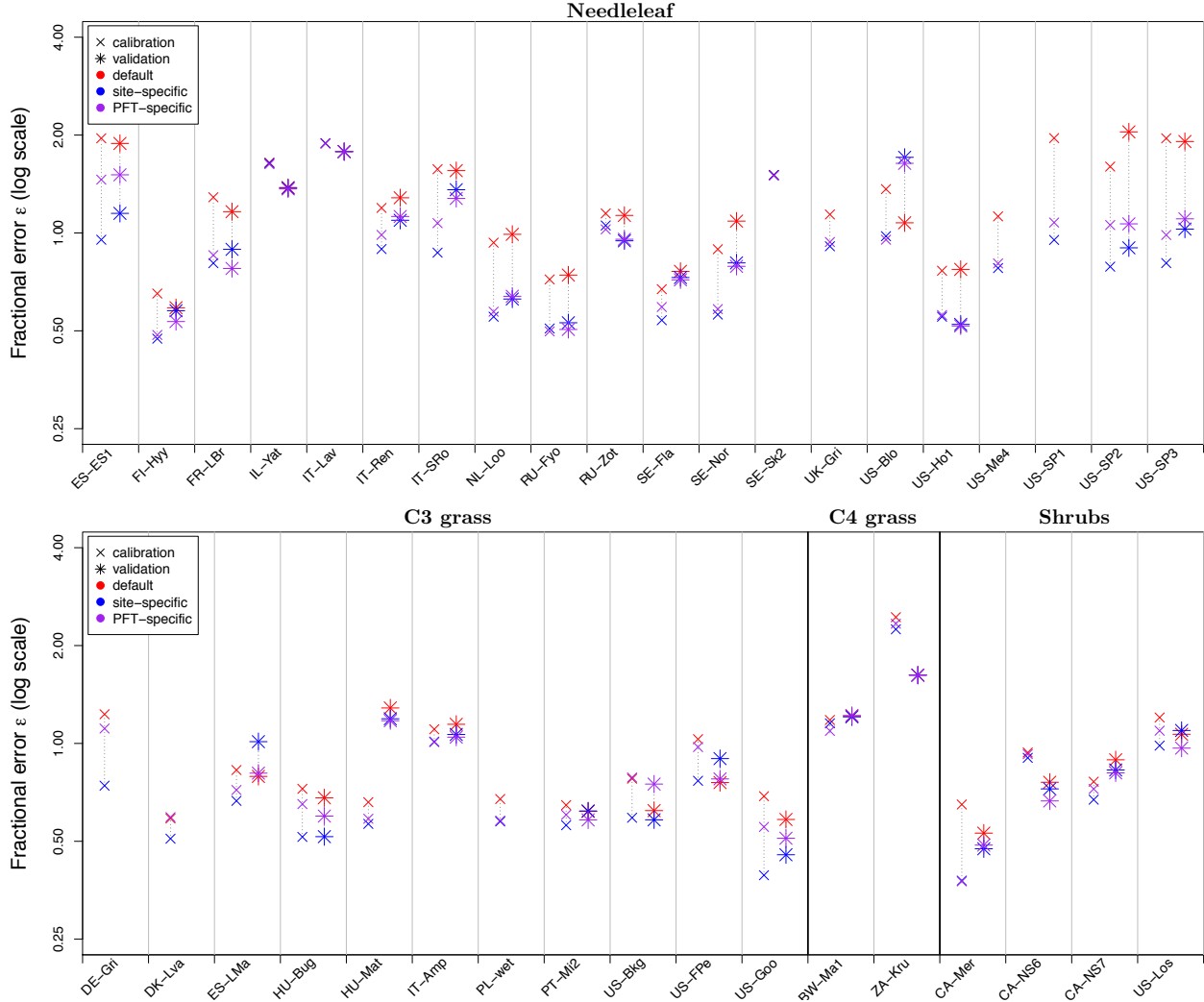

**Figure 5.** continued.

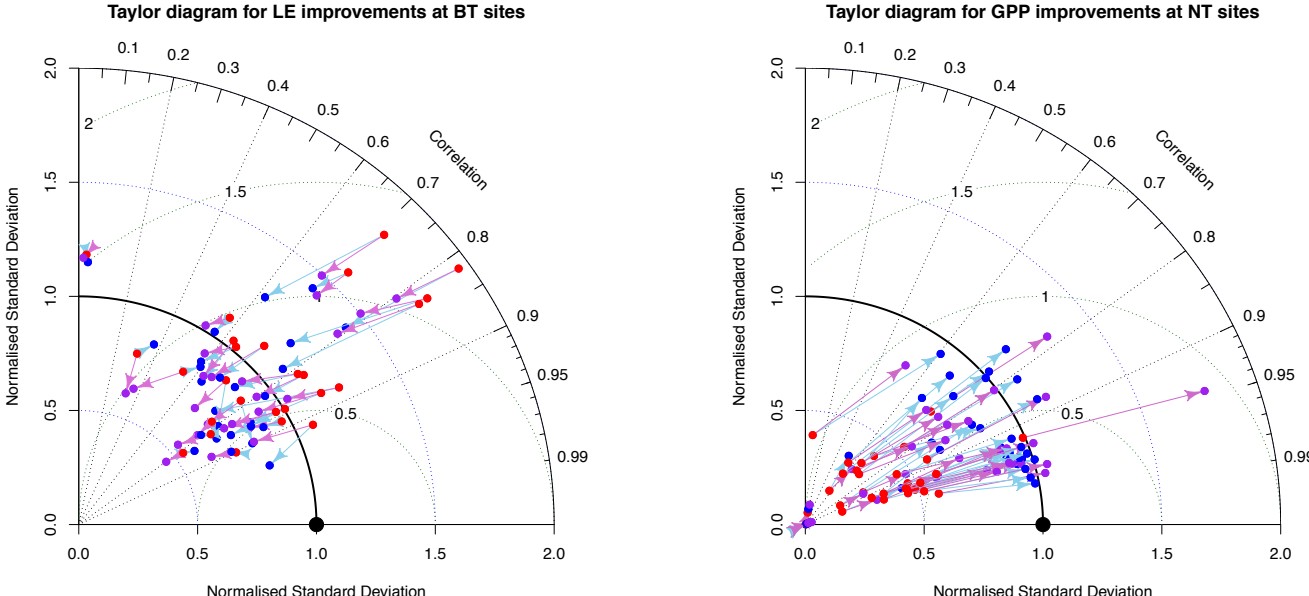

**Figure 6.** Improvements in fit represented by 'Taylor diagrams'. Observed timeseries (black dot) can be compared with modelled timeseries for default parameters (red dots), site–specific optimal parameters (blue dots) and PFT–generic optimal parameters (purple dots). Radial distance from the origin (dotted lines) represents normalised standard deviation $\sqrt{var(\mathbf{m}_t)/var(\mathbf{o}_t)}$, and so a modelled time series with the correct variance lies on the thick black line. Angular position represents the correlation between modelled and observed timeseries. The distance from the black dot (dotted green lines) represents the normalised standard deviation in the errors $\sqrt{var(\mathbf{o}_t - \mathbf{m}_t)/var(\mathbf{o}_t)}$.

**Table A1.** FLUXNET sites used in this study, labelled by a country code (first two letters) and site name (last three letters). The period corresponds to the available years of data for each of the sites.

| Site | Period | Calibration Year | Validation Year | Latitude | Longitude |
|---|---|---|---|---|---|
| **Broadleaf sites (BT)** | | | | | |
| DE-Hai | (2000,2006) | 2005 | 2004 | 51.079 | 10.452 |
| DK-Sor | (1996,2006) | 2006 | 2004 | 55.487 | 11.646 |
| FR-Fon | (2005,2006) | 2006 | - | 48.476 | 2.780 |
| FR-Hes | (1997,2006) | 2003 | 1998 | 48.674 | 7.065 |
| IT-Col | (1996,2006) | 2005 | 2001 | 41.849 | 13.588 |
| IT-LMa | (2003,2006) | 2006 | 2004 | 45.581 | 7.155 |
| IT-Non | (2001,2006) | 2002 | 2003 | 44.690 | 11.089 |
| IT-PT1 | (2002,2004) | 2003 | 2004 | 45.201 | 9.061 |
| IT-Ro1 | (2000,2006) | 2006 | 2005 | 42.408 | 11.930 |
| IT-Ro2 | (2002,2006) | 2004 | 2006 | 42.390 | 11.921 |
| UK-Ham | (2004,2005) | 2005 | - | 51.121 | -0.861 |
| UK-PL3 | (2005,2006) | 2006 | - | 51.450 | -1.267 |
| US-Bar | (2004,2005) | 2005 | - | 44.065 | -71.288 |
| US-Ha1 | (1991,2006) | 1996 | 1998 | 42.538 | -72.171 |
| US-MMS | (1999,2005) | 2002 | 2003 | 39.323 | -86.413 |
| US-MOz | (2004,2006) | 2006 | 2005 | 38.744 | -92.200 |
| US-UMB | (1999,2003) | 2003 | 2002 | 45.560 | -84.714 |
| US-WCr | (1999,2006) | 2005 | 2000 | 45.806 | -90.080 |
| AU-Tum | (2001,2006) | 2003 | 2005 | -35.656 | 148.152 |
| AU-Wac | (2005,2007) | 2006 | - | -37.429 | 145.187 |
| BR-Sa1 | (2002,2004) | 2003 | 2004 | -2.857 | -54.959 |
| BR-Sa3 | (2000,2003) | 2002 | 2003 | -3.018 | -54.971 |
| FR-Pue | (2000,2006) | 2006 | 2005 | 43.741 | 3.596 |
| ID-Pag | (2002,2003) | 2003 | - | 2.345 | 114.036 |
| IT-Cpz | (1997,2006) | 2004 | 2006 | 41.705 | 12.376 |
| IT-Lec | (2005,2006) | 2006 | - | 43.305 | 11.271 |
| PT-Esp | (2002,2004) | 2004 | 2003 | 38.639 | -8.602 |
| PT-Mi1 | (2003,2005) | 2005 | - | 38.541 | -8.000 |
| **C3 grasses sites (C3G)** | | | | | |
| DE-Gri | (2005,2006) | 2006 | - | 50.950 | 13.512 |
| DK-Lva | (2005,2006) | 2006 | - | 55.683 | 12.083 |
| ES-LMa | (2004,2006) | 2006 | 2005 | 39.941 | -5.773 |
| HU-Bug | (2002,2006) | 2006 | 2005 | 46.691 | 19.601 |
| HU-Mat | (2004,2006) | 2006 | 2005 | 47.847 | 19.726 |
| IT-Amp | (2002,2006) | 2006 | 2005 | 41.904 | 13.605 |
| PL-wet | (2004,2005) | 2005 | - | 52.762 | 16.309 |
| PT-Mi2 | (2004,2006) | 2006 | 2005 | 38.477 | -8.025 |
| US-Bkg | (2004,2006) | 2006 | 2005 | 44.345 | -96.836 |
| US-FPe | (2000,2006) | 2002 | 2004 | 48.308 | -105.101 |
| US-Goo | (2002,2006) | 2006 | 2004 | 34.250 | -89.970 |

| Site | Period | Calibration Year | Validation Year | Latitude | Longitude |
|------|--------|------------------|-----------------|----------|-----------|
| **Needleleaf sites (NT)** | | | | | |
| CA-Man | (1997,2003) | 2001 | 2002 | 55.880 | -98.481 |
| CA-NS1 | (2002,2005) | 2004 | 2003 | 55.879 | -98.484 |
| CA-NS2 | (2001,2005) | 2002 | 2004 | 55.906 | -98.525 |
| CA-NS3 | (2001,2005) | 2004 | 2002 | 55.912 | -98.382 |
| CA-NS4 | (2002,2004) | 2004 | 2003 | 55.912 | -98.382 |
| CA-NS5 | (2001,2005) | 2004 | 2002 | 55.863 | -98.485 |
| CA-Qcu | (2001,2006) | 2005 | 2006 | 49.267 | -74.037 |
| CA-Qfo | (2003,2006) | 2006 | 2005 | 49.693 | -74.342 |
| CA-SF1 | (2003,2005) | 2004 | 2005 | 54.485 | -105.818 |
| CA-SF2 | (2003,2005) | 2004 | 2005 | 54.254 | -105.878 |
| CA-SF3 | (2003,2005) | 2005 | 2004 | 54.092 | -106.005 |
| DE-Bay | (1996,1999) | 1999 | 1998 | 50.142 | 11.867 |
| DE-Har | (2005,2006) | 2006 | - | 47.934 | 7.601 |
| DE-Tha | (1996,2006) | 2005 | 2004 | 50.964 | 13.567 |
| DE-Wet | (2002,2006) | 2006 | 2004 | 50.453 | 11.457 |
| ES-ES1 | (1999,2006) | 2005 | 2000 | 39.346 | -0.319 |
| FI-Hyy | (1996,2006) | 2006 | 2004 | 61.847 | 24.295 |
| FR-LBr | (2003,2006) | 2006 | 2005 | 44.717 | -0.769 |
| IL-Yat | (2001,2006) | 2005 | 2006 | 31.345 | 35.051 |
| IT-Lav | (2000,2002) | 2001 | 2002 | 45.955 | 11.281 |
| IT-Ren | (1999,2006) | 2005 | 2006 | 46.588 | 11.435 |
| IT-SRo | (1999,2006) | 2006 | 2005 | 43.728 | 10.284 |
| NL-Loo | (1996,2006) | 2006 | 2003 | 52.168 | 5.744 |
| RU-Fyo | (1998,2006) | 2005 | 2006 | 56.462 | 32.924 |
| RU-Zot | (2002,2004) | 2003 | 2004 | 60.801 | 89.351 |
| SE-Fla | (1996,1998) | 1998 | 1997 | 64.113 | 19.457 |
| SE-Nor | (1996,1999) | 1997 | 1999 | 60.086 | 17.480 |
| SE-Sk2 | (2004,2005) | 2005 | - | 60.130 | 17.840 |
| UK-Gri | (1997,1998) | 1998 | - | 56.607 | -3.798 |
| US-Blo | (1997,2006) | 2006 | 2000 | 38.895 | -120.633 |
| US-Ho1 | (1996,2004) | 2004 | 2003 | 45.204 | -68.740 |
| US-Me4 | (1996,2000) | 2000 | - | 44.499 | -121.622 |
| US-SP1 | (2000,2001) | 2001 | - | 29.738 | -82.219 |
| US-SP2 | (1998,2004) | 2001 | 2004 | 29.765 | -82.245 |
| US-SP3 | (1999,2004) | 2001 | 2002 | 29.755 | -82.163 |
| **Shrubs sites (Sh)** | | | | | |
| CA-Mer | (1998,2005) | 2004 | 2005 | 45.409 | -75.519 |
| CA-NS6 | (2001,2005) | 2003 | 2004 | 55.917 | -98.964 |
| CA-NS7 | (2002,2005) | 2003 | 2004 | 56.636 | -99.948 |
| IT-Pia | (2002,2005) | 2003 | 2004 | 42.584 | 10.078 |
| US-Los | (2001,2005) | 2005 | 2003 | 46.083 | -89.979 |
| **C4 grasses sites (C4G)** | | | | | |
| BW-Ma1 | (1999,2001) | 2000 | 2001 | -19.916 | 23.561 |
| ZA-Kru | (2001,2003) | 2002 | 2003 | -25.020 | 31.497 |

**Table B1.** PFT-specific JULES parameters optimised in this study (Table 1). The prior values and ranges for each PFT are given. Below in bold are the optimised values and posterior uncertainty ranges given as an 80% confidence interval (in parentheses). Optimised values for which the prior values lie outside the posterior range are highlighted by (*). A graphical version of this table is shown in Figure 3.

| | BT | NT | C3 | C4 | Sh |
|---|---|---|---|---|---|
| $n_0$ | 0.046 | 0.033 | 0.073 | 0.06 | 0.06 |
| | (0.001,0.2) | (0.001,0.2) | (0.001,0.2) | (0.001,0.2) | (0.001,0.2) |
| | **0.061** | **0.065*** | **0.07** | **0.051*** | **0.041** |
| | **(0.034,0.066)** | **(0.059,0.07)** | **(0.018,0.145)** | **(0.043,0.056)** | **(0.006,0.066)** |
| $\alpha$ | 0.08 | 0.08 | 0.12 | 0.06 | 0.08 |
| | (0.001,0.999) | (0.001,0.999) | (0.001,0.999) | (0.001,0.999) | (0.001,0.999) |
| | **0.131*** | **0.096** | **0.179*** | **0.118*** | **0.102** |
| | **(0.087,0.14)** | **(0.021,0.167)** | **(0.155,0.209)** | **(0.075,0.141)** | **(0.063,0.763)** |
| $f_0$ | 0.875 | 0.875 | 0.9 | 0.8 | 0.9 |
| | (0.5,0.99) | (0.5,0.99) | (0.5,0.99) | (0.5,0.99) | (0.5,0.99) |
| | **0.765*** | **0.737*** | **0.817** | **0.765*** | **0.782*** |
| | **(0.655,0.787)** | **(0.713,0.758)** | **(0.727,0.944)** | **(0.752,0.793)** | **(0.735,0.848)** |
| $T_{\mathrm{low}}$ | 0 | -10 | 0 | 13 | 0 |
| | (-50,40) | (-50,40) | (-50,40) | (-50,40) | (-50,40) |
| | **1.203** | **-8.698** | **-1.985*** | **11.37** | **-5.208*** |
| | **(-0.555,9.492)** | **(-10.98,-6.342)** | **(-3.877,-0.13)** | **(7.522,14.072)** | **(-10.855,-2.106)** |
| $T_{\mathrm{upp}}$ | 36 | 26 | 36 | 45 | 36 |
| | (25,50) | (25,50) | (25,50) | (25,50) | (25,50) |
| | **38.578*** | **34.721*** | **36.242** | **44.897** | **35.385** |
| | **(38.157,40.698)** | **(33.214,36.365)** | **(33.087,38.599)** | **(44.201,46.426)** | **(26.339,40.216)** |
| $d_r$ | 3 | 1 | 0.5 | 0.5 | 0.5 |
| | (0.1,4) | (0.1,4) | (0.1,4) | (0.1,4) | (0.1,4) |
| | **3.009** | **1.425*** | **0.991*** | **0.404*** | **0.411*** |
| | **(2.901,3.052)** | **(1.159,1.672)** | **(0.901,1.101)** | **(0.5,3.623)** | **(0.324,0.473)** |
| $\frac{\delta c}{\delta l}$ | 0.05 | 0.05 | 0.05 | 0.05 | 0.05 |
| | (0.001,0.1) | (0.001,0.1) | (0.001,0.1) | (0.001,0.1) | (0.001,0.1) |
| | **0.047*** | **0.045*** | **0.05** | **0.05** | **0.048** |
| | **(0.046,0.049)** | **(0.042,0.048)** | **(0.047,0.052)** | **(0.046,0.054)** | **(0.04,0.055)** |
| $dq_c$ | 0.09 | 0.06 | 0.1 | 0.075 | 0.1 |
| | (0.001,0.2) | (0.001,0.2) | (0.001,0.2) | (0.001,0.2) | (0.001,0.2) |
| | **0.048** | **0.036** | **0.086** | **0.046*** | **0.077** |
| | **(0.02,0.183)** | **(0.008,0.066)** | **(0.07,0.109)** | **(0.045,0.053)** | **(0.024,0.118)** |