# Peer review of "Land surface parameter optimisation using data assimilation techniques: the adJULES system"

_Geoscientific Model Development, 2015_

## Referee Comment (RC1) · Anonymous Referee #1 · 24 Feb 2016

The manuscript "Land surface parameter optimisation through data assimilation: the adJULES system" by N. Raoult et al. presents a parameter calibration system for the JULES land surface model based on a 4D-Var approach employing the adjoint of JULES. The data to be assimilated are eddy covariance measurements of GPP and LE as provided by FLUXNET. The authors also compared the performance of single-site optimisation against multi-site optimisation for the 5 JULES PFTs. The multi-site optimisations (they call it 'PFT-specific') improves the fit against the assimilated data for over 90% of the sites used in the optimisation, and for about a third of the sites performs as well as the single-site optimisation.

The manuscript addresses an important topic in Earth System modelling, i.e. the objective calibration of model parameters against observations and reduction of uncertainty for the land surface component of the UK Met Office climate and weather forecast

models. Such objective calibration systems for Earth System Model land surface components are still in their infancy, and thus not routinely performed, which is one of the reasons for the wide spread in the latest climate model intercomparison results especially for the terrestrial carbon cycle components. Hence, this contribution is very timely. Unfortunately, the manuscript suffers from many inconsistencies and lack of precision which makes it sometimes a rather awkward read. The manuscript would have certainly benefited from careful proof reading by all (co-)authors before submission.

One example of this is the authors' use of the term 'adJULES'. Sometimes they refer with adJULES to the adjoint of JULES and sometimes to the whole optimisation system. The two are certainly very different and as such should also be clearly distinguished in the manuscript. There is an established terminology in the data assimilation community and it would improve the readability if the authors would use this terminology, e.g. 'posterior' instead of 'new' parameter.

The authors claim that any residual differences between the observations and model output using the optimised parameter vector are due to structural errors in the model and not to the parameter values. This may be true if they have really identified the best possible fit, i.e. if they have found the global cost function minimum. Since with such complex models the cost function usually has a multimodal structure it is not clear that a gradient-based optimisation approach finds the global minimum. The authors need to comment on that in the manuscript. In fact, the manuscript would benefit from including some posterior diagnostics, such as the final cost function and gradient values. It is not clear if they've always found a minimum, and if so if that is the global minimum.

The study also lacks some independent validation. The authors only calculate the improvement in RMSE for the same data streams they also assimilate. A careful validation against independent data is especially important because by calibrating the model parameters against a specific data set the model's performance may be deteriorated compared to other independent data.

**GMDD**
Specific comments: P3 L3: The term 'adJULES' should be defined before using it.

P4 Eq 1: The cost function is missing the factor  $\frac{1}{2}$ . The omission of this factor in the calculations leads to a wrong estimation of the posterior uncertainties.

P4 L17: What do you mean by 'observed covariance in the error (m-o)'? How can you observe this?

P4 L19: How does lambda enter Eq 1?

P4 L28/29: This sentence needs to be reformulated. It is not clear how reverse and forward mode relate to the adjoint. The adjoint calculates the derivative in reverse mode.

P5 Fig 1: Essentially the figure is incomprehensive and does not show an interative loop.

P6 L1: The data selection criteria should be specified exactly. What does 'significant gaps' mean. There is also the danger of introducing biasas by certain data selection criteria. This should be taken into account.

P6 L3: Why does one require NEE and LE fluxes to model photosynthesis? Please clarify.

P6 L5: The eddy covariance technique measures the net exchange flux and not GPP. The net flux is partitioned into GPP and respiration by a model. So essentially, in this study the authors calibrate the JULES model against another model, which is used to obtain GPP from eddy covariance measurements. This needs to be discussed.

P6 L6/7: This procedure may lead to inconsistencies between the actual vegetation at a given site and the vegetation structure and soil type used in the model. This should be discussed in the manuscript.

P6 L8: Please provide a reference for the LAI product. Here again, this may lead to another inconsistency, see point above.

GMDD
P6 L31/32: Please rephrase. The adjoint does not find the second derivative.

P6 L33: How did you determine the weights? What do you mean by 'low enough'?

P7 Ll11-17: This is an interesting way to calculate the posterior parameter uncertainties, but it is not clear why and what exactly you do there. What is the advantage of using this method over calculating the posterior uncertainties from the inverse of the Hessian directly? When you calculate the full Hessian you also get the full error covariance matrix! Do you get a semi-definite Hessian (see also general comment on obtaining a minimum)?

Sect 2.5.2: What is the advantage of the metric you define here over calculating the relative uncertainty reduction with respect to the prior? This also provides and assessment of the quality of the fit and is a common diagnostic in data assimilation. It is also not clear how a complete mismatch looks like.

P8 L5: This is not a validation, but rather an assessment of the how good the fit against the data is. A real validation would be against independent data and not the data used for assimilation.

P8 L25: Why does JULES not perform very well for C4 grasses. You should elaborate this.

P8 L 31: What do you mean by the 'adjoint performs well'? Does it perform well in terms of efficiency? And if so, how efficient is the adjoint?

P9 Fig 2: Which sites are you showing and what are the units? On what basis did you select the shown sites?

P9 last sentence: Why didn't you include these parameters in the optimisation?

P10 Fig 2 caption: Please remove the extra 'vector'.

P10 L2: Again, 'validate' is the wrong word here. And why only for broadleaf sites?
P10 L4: The sentence need to rephrased.

P10 L6: What do you mean by 'training sets'? This sounds a bit like as if you were using a neural network approach, which has to be trained.

P10 L8: What are these sets?

P11 L1-3: Why should adding more sites render the cost function more smoothly? It could also be the opposite, please explain in the manuscript.

P11 Sec 3.3: This section is really only a description of the posterior parameters but they need to be discussed as well and put in context of a) their prior values, b) their physical meaning and c) the covariances with respect to the resulting fluxes and a successful optimision.

P11 L9: What do you mean by 'new uncertainties'?

P11 L12/13: The uncertainties cannot be skewed, it's the PDF that can be skewed.

P11 L15: What is the 80% conficence interval, how did you calculate this?

P11 L30/31: Why are the correlations related to the number of sites used in the optimisation? Please explain in the manuscript.

P12 L10/11: What makes the UK-PL3 site different? Please explain in the manuscript.

P12 Sec 3.3.3: As mentioned in the general comments, the calibrated parameter set should be evaluated against independent data.

P12 L25/26: What do you mean here? Please rephrase the sentence.

Fig 4: Please label the rows. Maybe increase the bar size to improve readability.

Fig 6: What is the difference between top and bottom panel and what to the vertical lines denote? What are the outliers that have been removed and why did you remove them?
**GMDD**

---

## Referee Comment (RC2) · Anonymous Referee #2 · 23 Mar 2016

Review of paper "Land surface parameter optimisation through data assimilation: the adJULES system by Raoult et al.

Thank you for inviting me to review this paper.

Data Assimilation is often talked about in meteorological/climate modelling, and indeed is used in things like the ECMWF re-analysis products. But this is one of the first applications to land surface modelling, and so this paper is particularly important.

The reference list is comprehensive and useful, alongside good illustrative diagrams and a useful table across the FLUXNET sites.

The paper should be published in GMD, and below are a few suggestions the authors might like to consider in any final refined version. It's a long list of many points, some

almost trivial – if the paper was printed in its current form, that would probably be OK. But there may be a few things below that if the authors adopt, may just sharpen the document.

I am happy to see the paper again as a reviewer.

General things

The Abstract is clear, although just reading the Abstract, a question might be asked as to why the data is not split in to training and test data?

Maybe expand just slightly on "a third of which give similar reduction in errors as site specific optimisations". The point being made here is that this suggests parameters are similar and robust between sites. This is always good news for climate modelling, suggesting it is possible to reduce to relatively small numbers of PFTs. Maybe stress this point a bit more? (However, if this is stressed more, then need to explain Groenendijk et al 2010?).

Lines 34, page 2 – Lines 3, page 3. This feels as if it undersells the adjoint approach! I would make a key bullet point that this is a more sophisticated approach (via matrix inversion) to finding rapidly minima across multiple parameters. It would be almost impossible to replicate these findings using some sort of brute-force optimisation, with nested loops over different parameters.

Around Eqn (1), line 11. Sentence "A cost function f(z)…" looks like it has remained in by accident, and then the correct sentence is the next one. "The cost consists…." (The second sentence correctly identifies that the z-z0 differences also contribute to cost function in Eqn(1)).

Eqn(1) – Has Lambda been accidently dropped from Eqn (1) . It should multiple the second term? (I realise line 21 states it is taken as unity, but I'd still put it in Eqn(1), and state line 21 "All parameters and observations are equally weighted in this cost function – i.e. lambda=1"

Is there a good reason for selecting lambda=1 (or its implications)? Does it imply we put equal trust in the FLXUNET measurements (left term) as the local measurements that give the local parameters (right term). A couple of words on this might help the reader?

Possibly me being confused, but if B is a diagonal matrix, then this isn't about co-variances – which imply off-diagonal terms?

Section "Multisite Implementation". Could tighten slightly to say something like "and this would introduce a double summation in Eqn(1), over n locations. Hence R and B become matrices of size [n*s x n*s]". Is that correct? This would fit with, as stated, to find "values for a common set of parameters". This gives single values for each z parameter. The wording of the last sentence is slightly ambiguous? "Similarly, the first and second derivative. . ...using the sum of the derivatives at the individual sites". This reads as if the derivatives are calculated locally, and then a mean taken. Would in fact a single sweep across all n*s data points be used, and the derivatives calculated once, if common parameters are investigated. [Maybe eqn term cancellation implies they are the same, but. . ...?].

Somewhere in Section 2.3 or 2.4 – possibly remind readers that FLUXNET also comes with the meteorological data. (In other words, it's not just the fluxes and then something like NCEP or ECMWF data was used additionally to give met drivers).

Section 2.5.2. Would need to be confident that outlier points didn't do something odd in Eqn (4)? I guess the initial sweep of data ensures this is OK? (The alternative would be to normalise with SD of (mi,t)), and then get the percentage of variance explained).

Figure 3. This is a great figure. However, pictures and captions often get pulled out of papers and shown in isolation. To ensure information is safely contained, mention in caption (or across top of plot), these are broadleaf trees only? At first glance, I thought the y-axis was some sort of physical unit (for LE or NPP). However caption says this is from the 2.5.2 Section metric. The normalised values are very small, and don't ever get

near unity. Could this be the outliers mentioned above? Not a problem, but bottom of page 7 says "1 – a complete mismatch" Wouldn't we expect some of the parameters to perform quite badly, and get a bit nearer to unity? Or - does this mean that in general, even without parameter fitting, then JULES is an exceptionally good model? Fitting reduces that last small error down further?

Figure 3 mentions training versus validation. This appears different to the impression of the Abstract that all data is used to train?

Figure 3. Usual practice is to put the legend inside the plot – there is space for the 5 symbols, top left hand maybe?

Figure 4 is great. But on my print out, the vertical lines cannot be seen in many instances. Thicken them maybe? As always, a matter of style, so just a suggestion. To make Figure 4 less crowded, would it make sense to not put the value of original & optimised as text annotations as this repeats information in the plots. Then the plots can be made bigger and bolder? Maybe put units in left column?

Section 3.3 and Figure 5. What is so remarkable about Figure 5 is that the strong correlations between parameters are not consistent across the PFTs. Maybe not for this paper, but some sort of physical interpretation of that would be really interesting. Returning to the governing equations and their scaled amounts might help. Is "correlation" the best word – "collinearity" might be more appropriate?

Figure 6 is like Figure 3, but a lot less cluttered. The data on Figure 6, BTs is same as that on Fig 3, except the "multi-site". Looking back at Fig 3, need to understand better the "five sites" algorithm (again, page 10, line 4 – some text accidently deleted?).

Figure 6 – I'd make the lower y-axis bound 0.0 (rather than what looks like 0.001)? Gives a better feel then of the improvement in absolute terms.

Conclusions – To my eye, Figure 6 says it all, and I would stress far more the real headline findings that:

(!) There is a general reduction in error of around 50%

(2) Possibly of more importance, using cross-PFT parameters, often get very similar improvements than local fits. This implies robust parameterisations independent of geography – which GCM modellers always like to see.

Small things

Maybe get the words "Data Assimilation" used a few times on the paper on page 1 / Abstract, so it gets picked up for anyone using that expression in an Internet search. (It's an older terminology used for this sort of approach, but is still valid).

Abstract: Line 2, maybe mention that JULES is also used comprehensively as an impacts tool, sometimes forced with known climatologies and/or alternative GCMs in to the future. So it is not used just coupled to UK Met Office models.

Abstract - could "automatically differentiated" be expanded slightly to "automatically differentiated with respect to JULES parameters......"

Maybe line 25, page 1. To make topical post COP21, could add something like: "Any future decreased ability of the land surface to draw-down atmospheric CO2 could imply fewer "permissible emissions" in order to stay below key warming thresholds such as two degrees"

Top page 2. Is there a process other than nitrogen cycling that can be mentioned – preferably one that has been introduced in to the JULES model version used here?

Sentence "Given the small spatial footprint....". Maybe clarify why this gives over-tuning? Presumably because it might be see some anomalous plants in the small footprint, and that are not representative of PFTs over a broader area?

Bottom of page 3. Line 30. Could mention that "available observations" are about independent large-scale measurements (such as FLUXNET)? These are different to the specific process measurements used to calibrate the individual components that

are mentioned in line 27.

Page 4, line 5. Possibly: "As used widely in weather forecasting, along with other disciplines".

Page 4, line 14. Maybe: " ….outputs at time 1 <= t <= s" (so defines s).

Page 4, line 22. To anyone new to data assimilation, could say: "optimal vector….minimizes the cost function (Eq. 1) via JULES model itself though m=m(z) (left terms) and directly via z in the right terms"

Page 4, line 25. I can understand box constraints = upper or lower bounds. But what does "limited memory" refer to?

On the diagram, Figure 1, right-hand side, maybe word as "Hessian to give uncertainty bounds"

Page 5, line 14. Table A1 has a lot fewer than 500 entries, so quite a lot of data is rejected?

Page 7, line15. "One or two" refers to whether a plot is a standard plot, or a contour plot in two parameters?

Page 8, line 5. The analysis here is more testing the concept of common parameters between sites, rather than testing the methodology?

Figure 2 is really nice. Just a few small things. Is there are reason black is also dashed? Style thing, but I've have maybe put as the individual panel titles "Broadleaf LE, Broadleaf GPP", etc. So across the top of the panels. And then the y-axes, put the units – so W/m2 left panels etc. Then possibly not bothered with the labels (a)-(e)? Maybe make the lines with slightly larger line width? Inside each box, give the site ID as annotated text, as these are timeseries for just single sites.

Page 10, line 4. Some text missing from sentence?

---

## Author Response (AR1)

**Response to Reviewer #1:**

The authors would like to thank Reviewer #1 for taking the time to write such helpful, thorough and constructive comments. The comments have been taken into consideration in the revised manuscript. We answer them individually as follows:

**1 General comments:**

**Sometimes they refer with adJULES to the adjoint of JULES and sometimes to the whole optimisation system. The two are certainly very different and as such should also be clearly distinguished in the manuscript.**

This has now been clarified, adJULES is used to refer to the whole optimisation scheme. The bracket containing 'called adJULES' was removed from page 4, line 6, which we believe was the source of this confusion.

**There is an established terminology in the data assimilation community and it would improve the readability if the authors would use this terminology, e.g. 'posterior' instead of 'new' parameter.**

This has been addressed, the text has been changed in places where established terminology would improve readability, for example in section 3.2.1 'Assessment of PFT-specific optimal parameters'.

**The authors claim that any residual differences between the observations and model output using the optimised parameter vector are due to structural errors in the model and not to the parameter values. This may be true if they have really identified the best possible fit, i.e. if they have found the global cost function minimum. Since with such complex models the cost function usually has a multimodal structure it is not clear that a gradient-based optimisation approach finds the global minimum. The authors need to comment on that in the manuscript. In fact, the manuscript would benefit from including some posterior diagnostics, such as the final cost function and gradient values. It is not clear if they've always found a minimum, and if so if that is the global minimum.**

It is true that the limitations of a gradient-based optimisation approach is missing from the manuscript. The following text has been added to the conclusion to address this omission and to reduce the emphasis placed on model structure errors

*" A limitation of gradient descent methods, such as the optimisation scheme used in this study, is the fact that sometimes a local minimum in found instead of the global minimum. However, as discussed in section [..] Kuppel et al (2014)'s hypothesis that the cost function becomes smoother with additional sites may be a solution in avoiding local minima. Alternative methods, including ensemble methods, could avoid this issue, but are more computationally costly."*

**The study also lacks some independent validation. The authors only calculate the improvement in RMSE for the same data streams they also assimilate. A careful validation against independent data is especially important because by calibrating the model parameters against a specific data set the model's performance may be deteriorated compared to other independent data.**

Given the small number of sites available to us, we decided to use all available sites in finding the multisite parameters sets. Sites used in this study required at least two consecutive years, one to spin-up the model and one to calibrate against. Re-examining the data, we found that the majority of

the sites had more than two years and so a different year could be used to validate the optimised set of parameters. The year used for validation was chosen to be the second most complete, the first most complete having been used in the calibration. The results of the validation, which are very positive, are now shown in the results section alongside the results for the calibration.

**2 Specific comments:**

**P3 L3: The term 'adJULES' should be defined before using it.**

Corrected: now defined on P2 L31.

**P4 Eq 1: The cost function is missing the factor 1/2. The omission of this factor in the calculations leads to a wrong estimation of the posterior uncertainties.**

Factor added.

**P4 L17: What do you mean by 'observed covariance in the error (m-o)'? How can you observe this?**

Removed "observed" from the sentence.

**P4 L19: How does lambda enter Eq 1?**

An explanation has been added.

**P4 L28/29: This sentence needs to be reformulated. It is not clear how reverse and forward mode relate to the adjoint. The adjoint calculates the derivative in reverse mode.**

Removed part of the sentence: "in 'reverse mode' (rather than 'forward mode') for computational efficiency", and the following text was added: *"Automatic differentiation relies on using the chain rule, the choice of forward or reverse mode refers to the order in which the derivatives are computed."*

**P5 Fig 1: Essentially the figure is incomprehensive and does not show an interative loop.**

There are two iterative loops in our system, one found within the minimisation scheme itself (BFGS) and one created by re-feeding zin the system. This second loop is need since the covariance matrix $\mathbf{R}$ is dependent on $\vec{z}$. This fact has now been explained more explicitly in the text. Eq(1) now reads:

$$f\left(\vec{z}; \hat{\vec{z}}, \vec{z}_0\right) = \frac{1}{2}\left[\sum_t (\mathbf{m}_t(\vec{z}) - \mathbf{o}_t)^T \mathbf{R}\left(\hat{\vec{z}}\right)^{-1}(\mathbf{m}_t(\vec{z}) - \mathbf{o}_t) + \lambda(\vec{z} - \vec{z}_0)^T\mathbf{B}^{-1}(\vec{z} - \vec{z}_0)\right]. \qquad (1)$$

*Here, $\mathbf{R}(\hat{\vec{z}}) = \frac{1}{n}\sum_{t=1}^{n}(\mathbf{m}(\hat{\vec{z}})_t - \mathbf{o}_t)(\mathbf{m}(\hat{\vec{z}})_t - \mathbf{o}_t)^T$ denotes the error cross product matrix produced by a JULES run with parameter value $\hat{\vec{z}}$. In an optimisation, $\vec{z}$ and $\hat{\vec{z}}$ are updated separately in nested loops, having both been initialised to the default JULES parameter value $\vec{z}_0$. In the inner loop, $\vec{z}$ is varied to minimise the cost function (termination criterion: $\nabla f \approx 0$) for the current value of $\hat{\vec{z}}$. In the outer loop, $\hat{\vec{z}}$ is reset to the new value of $\vec{z}$ from the inner loop (termination criterion: change in $\hat{\vec{z}}$ negligible). At the end of an optimisation, therefore, the matrix $\mathbf{R}$ conveys information about the error correlation structure in a JULES run with optimal parameter values.*

The figure has also been amended, removing the criterion $\nabla f \approx 0$ from the question box, since it was incorrectly referring to the BFGS terminating condition and not the z terminating condition.

**P6 L1: The data selection criteria should be specified exactly. What does 'significant**

gaps' mean. There is also the danger of introducing biases by certain data selection criteria. This should be taken into account.

Sites with data gaps of more than 50% during the growing season or missing input variables were excluded from the analysis. This has been clarified in the text replacing "significant gaps" with "data gaps of more than 50%".

**P6 L3: Why does one require NEE and LE fluxes to model photosynthesis? Please clarify.**

Sentence rephrased: "To constrain photosynthetic parameters, Net Ecosystem Exchange (NEE) and Latent Heat flux (LE), among other fluxes, are helpful."

**P6 L5: The eddy covariance technique measures the net exchange flux and not GPP. The net flux is partitioned into GPP and respiration by a model. So essentially, in this study the authors calibrate the JULES model against another model, which is used to obtain GPP from eddy covariance measurements. This needs to be discussed.**

Text added: *"GPP data are model-derived estimates, which could introduce an additional uncertainty into the results. This is kept in mind during the analysis."*

**P6 L6/7: This procedure may lead to inconsistencies between the actual vegetation at a given site and the vegetation structure and soil type used in the model. This should be discussed in the manuscript.**

Sentence added: *"This could lead to inconsistencies between the actual vegetation at a given site and the vegetation structure and soil type used in the model. This is kept in mind during the analysis."*

**P6 L8: Please provide a reference for the LAI product. Here again, this may lead to another inconsistency, see point above.**

Reference for MODIS data added: *"Myneni, R.B., Hoffman, S., Knyazikhin, Y., Privette, J.L., Glassy, J., Tian, Y., Wang, Y., Song, X., Zhang, Y., Smith, G.R. and Lotsch, A., 2002. Global products of vegetation leaf area and fraction absorbed PAR from year one of MODIS data. Remote sensing of environment, 83(1), pp.214-231."*

**L31/32: Please rephrase. The adjoint does not find the second derivative.**

Rephrased: *"The second derivative of the cost function found by differentiation of the adjoint code..."*

**P6 L33: How did you determine the weights? What do you mean by 'low enough'?**

Text added to the Experiment setup section, explaining the tuning of lambda for the multisite cases:

*"Preliminary experiments showed very narrow uncertainties whilst running the optimisation scheme over multiple sites i.e. the background term was found to dominate the cost function. In previous multisite studies (Kuppel et al., 2012, 2014), the prior range was also used to defined the background covariance matrix B. The range was variously further multiplied by a factor of 40% (Kuppel et al., 2012) and 1/6 (Kuppel et al., 2014). Experiments were run to find a similar factor to use in this study (the constant of proportionality in Eq. 5). In each of the multisite experiments, the lowest value of such that the Hessian is positive definite at the optimal parameter value was used. This allows uncertainties to be generated around each parameter and prevents the gradient descent algorithm from reaching the boundaries of the prescribed prior range."*

**P7 Ll11-17: This is an interesting way to calculate the posterior parameter uncertainties, but it is not clear why and what exactly you do there. What is the advantage of using**

this method over calculating the posterior uncertainties from the inverse of the Hessian directly? When you calculate the full Hessian you also get the full error covariance matrix! Do you get a semi-definite Hessian (see also general comment on obtaining a minimum)?

The adJULES system is run using box constraints on the prior, giving it a (multivariate) top-hat distribution. The methodology was picked due to the fact the posterior PDFs will be truncated multivariate normal distributions due to the prescribed prior bounds given to each of the parameters. Text added on L11 to clarify this:

*"... Hessian is used to generate samples from the posterior distribution. This is a truncated multivariate normal distribution because of the box constraints placed on the prior."*

**Sect 2.5.2: What is the advantage of the metric you define here over calculating the relative uncertainty reduction with respect to the prior? This also provides and assessment of the quality of the fit and is a common diagnostic in data assimilation. It is also not clear how a complete mismatch looks like.**

This metric was chosen because not only does it show the improvement made by the optimised parameter vectors but could also be used to see how different sites performed compared to each other. The metric has been amended slightly to define the fraction of variance unexplained, which is more intuitive. Paragraph added to this effect on line 19:

*"This metric was chosen to show not only the improvement made by the optimal parameter vectors at each site but also to show how each site performed relative to others."*

**P8 L5: This is not a validation, but rather an assessment of the how good the fit against the data is. A real validation would be against independent data and not the data used for assimilation.**

The purpose of section 3.2 was to shows that given a set of 5 randomly selected sites, the optimised parameter vector found by optimising over these sites also improves the rest of the sites not used in the calibration. This experiment is now obsolete since we have the ability to validate the PFT-specific parameters properly in our improved result section. As a result, this section has been removed.

**P8 L25: Why does JULES not perform very well for C4 grasses. You should elaborate this.**

Text added P8 L26 to the effect: *"The original stomatal conductance-photosynthesis model within JULES was developed based on fluxes measured over C4 grass as part of the FIFE field experiment (Cox and Huntingford, 1998). However, there are relatively few Fluxnet sites over C4-dominated landscapes, and only two even in the extended dataset that we use. As a result ,the sensitivity of stomatal conductance and photosynthesis to environmental factors has been less well tested for C4 grasses. The results presented in this paper therefore highlight the need to reassess JULES and other land-surface models for predominantly C4 landscapes."*

**P8 L 31: What do you mean by the 'adjoint performs well'? Does it perform well in terms of efficiency? And if so, how efficient is the adjoint?**

Sentence changed to: *"the adJULES system works well in finding optimal parameter vectors which improve the performance of JULES at individual sites, regardless of PFT"*.

**P9 Fig 2: Which sites are you showing and what are the units? On what basis did you select the shown sites?**

The site identification code has been added to the plots and the units moved from the top of the figure

to the side of each individual panel for clarity. The sites picked were the ones that captured best the general trends for each of the PFTs. This was done manually.

**P9 last sentence: Why didn't you include these parameters in the optimisation?**

As our focus is on the carbon cycle, we choose to only optimised parameters directly relating to the photosynthesis equations in the JULES model. Given more time and more computing power, more parameters could be used in the optimisation.

**P10 Fig 2 caption: Please remove the extra 'vector'.**

Corrected.

**P10 L2: Again, 'validate' is the wrong word here. And why only for broadleaf sites?**

With the removal of section 3.2, this is no longer relevant.

**P10 L4: The sentence need to rephrased.**

Similarly, this sentence was removed when section 3.2 was suppressed.

**P10 L6: What do you mean by 'training sets'? This sounds a bit like as if you were using a neural network approach, which has to be trained.**

No longer relevant with the removal of section 3.2.

**P10 L8: What are these sets?**

No longer relevant with the removal of section 3.2.

**P11 L1-3: Why should adding more sites render the cost function more smoothly? It could also be the opposite, please explain in the manuscript.**

With the removal of section 3.2, this is no longer relevant. However, this is a phenomenon hypothesised in Kuppel et al. (2014) and there are a couple of examples of this happening in figure 6 (old). As a result, the text on P11 L1-3 has been suppressed and the following text added to page 21:

*"For some sites, US-Blo and BW-Ma1 for example, the PFT-generic parameter vector over-performs the parameter vector found locally. This phenomenon was also noted in Kuppel et al. (2014). The study further suggested that the added simultaneous constraints placed on the parameters by increasing the number of sites used in the cost function caused the cost function to become 'smoother' and so the optimisation scheme is less likely to get stuck in local minima."*

**P11 Sec 3.3: This section is really only a description of the posterior parameters but they need to be discussed as well and put in context of a) their prior values, b) their physical meaning and c) the covariances with respect to the resulting fluxes and a successful optimisation.**

This section has been expanded to include a more thorough analysis of the correlations in the context of physical interpretation. In order to achieve this, a description of the relevant JULES equations has been added to section 2.1. This puts the parameters in terms of the equations they govern. These equations are then used in section 3.2 to explain why some of the parameters vary the way they do.

**P11 L9: What do you mean by 'new uncertainties'?**

Sentence changed to: "the prior parameter value is found outside the posterior uncertainty bounds".

**P11 L12/13: The uncertainties cannot be skewed, it's the PDF that can be skewed.**

Corrected.

**P11 L15: What is the 80% confidence interval, how did you calculate this?**

The 80% fraction interval was calculated by taking the difference between the 90th and 10th quantile and dividing by the prescribed range. To make this clearer, this interval has been renamed the 80% "quantile" interval. The following description is added to section 2.5.1:

*"In order to illustrate the parameter uncertainties, error bars are used to represent the 80% quantile range (10th to 90th percentile) for each optimal parameter."*

**P11 L30/31: Why are the correlations related to the number of sites used in the optimisation? Please explain in the manuscript.**

This was a trend that was observed - the more sites used in the calibration, the more pronounced the correlations seemed to be. However, there was not enough time to run this experiment fully for the paper. The higher correlations found between the parameters for the BT and NT compares to the grass PFTs has now been address in the response to comment P11 Sec 3.3. As a results, this hypothesis has been removed from the text.

**P12 L10/11: What makes the UK-PL3 site different? Please explain in the manuscript.**

Text added: *"This UK site is in the Pang/Lambourn catchment, which has chalk soil with macropores that permit significant lateral subsurface flows of soil moisture. These horizontal flows cannot be captured in a model like JULES which is essentially one-dimensional in the vertical below the soil surface."*

**P12 Sec 3.3.3: As mentioned in the general comments, the calibrated parameter set should be evaluated against independent data.**

Issue addressed in response to general comments.

**P12 L25/26: What do you mean here? Please rephrase the sentence.**

The conclusion was been reworded due to the addition of validation to our study.

**Fig 4: Please label the rows. Maybe increase the bar size to improve readability.**

We have been experiencing issues with some printed version of the PDF, sometimes some of the information is missing. We will address this. On the online version, the rows are labelled with each parameter symbol. In order to declutter this figure, the prior and posterior values have been removed since this values are made explicit in Table B1. This has allowed us to increase the error bar plots. The lines have been increased to improve readability.

**Fig 6: What is the difference between top and bottom panel and what to the vertical lines denote? What are the outliers that have been removed and why did you remove them?**

As mentioned above, printed versions of this paper may show incomplete plots, the online version should still contain all the information. The two panels are the same, with Broadleafs and C3 grasses shown in the top panel, and Needleleafs, Shrubs and C4 grasses shown in the bottom panel. The vertical lines are there to break up the different PFTs. The outliers were removed from the plot because they made plot unreadable with much higher errors than the rest in the plot (x10). The sites removed are listed in the caption to the figure. With the slightly adjusted metric, as discussed previously, there are now only 2 such outliers and the data has now been split into 4 panels. An "i.e."

was added into the brackets to clarify that this list contains the outliers.

This is now discussed more thoroughly the conclusion.

**Lines 34, page 2 - Lines 3, page 3. This feels as if it undersells the adjoint approach! I would make a key bullet point that this is a more sophisticated approach (via matrix inversion) to finding rapidly minima across multiple parameters. It would be almost impossible to replicate these findings using some sort of brute-force optimisation, with nested loops over different parameters.**

Text has been added on line 33 to this effect: *"The adJULES system uses the adjoint method which finds minima rapidly across multiple parameters via matrix inversion and has the advantage of reproducibility. Replicating these findings using brute-force optimisation would be prohibitively expensive computationally."*

**Around Eqn (1), line 11. Sentence "A cost function f(z). . ." looks like it has remained in by accident, and then the correct sentence is the next one. "The cost consists. . .." (The second sentence correctly identifies that the z-z0 differences also contribute to cost function in Eqn(1)).**

Sentence starting with *"A cost function $f(z)...$"* has been removed and the second sentence rephrased as follows: *"The cost function, $f(\vec{z})$, consists of a weighted sum of squares of the difference between $\vec{m}_t$ (the vector of model outputs at time t), and $\vec{o}_t$ (the vector of observations at time t), combined with a term quadratic in the difference between parameter values $\vec{z}$ and initial parameter values $\vec{z}_0$"*

**Eqn(1) - Has Lambda been accidentally dropped from Eqn (1) . It should multiple the second term? (I realise line 21 states it is taken as unity, but I?d still put it in Eqn(1), and state line 21 "All parameters and observations are equally weighted in this cost**

**function - i.e. lambda=1"**

Lambda added to equation.

**Is there a good reason for selecting lambda=1 (or its implications)? Does it imply we put equal trust in the FLXUNET measurements (left term) as the local measurements that give the local parameters (right term). A couple of words on this might help the reader?**

The cost function was set up such that both terms are equally weighted. This is used in the single site experiment. Due to the number of sites, it was not possible to tune the value of $\lambda$ for each individual site. For the multisite experiments, more time was spent tuning this value. What value $\lambda$ should have is still something we need to look at, hopefully we will get a chance to understand it properly in further study. Text added to the Experiment setup section, explaining the tuning of $\lambda$ for the multisite cases:

*"Preliminary experiments showed very narrow uncertainties whilst running the optimisation scheme over multiple sites i.e. the background term was found to dominate the cost function. In previous multisite studies (Kuppel et al., 2012, 2014), the prior range was also used to defined the background covariance matrix B. The range was variously further multiplied by a factor of 40% (Kuppel et al., 2012) and 1/6 (Kuppel et al., 2014). Experiments were run to find a similar factor to use in this study (the constant of proportionality in Eq. 5). In each of the multisite experiments, the lowest value of such that the Hessian is positive definite at the optimal parameter value was used. This allows uncertainties to be generated around each parameter and prevents the gradient descent algorithm from reaching the boundaries of the prescribed prior range."*

**Possibly me being confused, but if B is a diagonal matrix, then this isn't about covariances? which imply off-diagonal terms?**

The parameters are assumed to start off uncorrelated. Text added: *"...The matrix $\mathbf{B}$ describes the prior covariances assigned to the parameters, and is here chosen to be a diagonal matrix proportional to the inverse square of the ranges allowed for each parameter. The prior uncertainties are therefore assumed to be uncorrelated between the parameters."*

**Section "Multisite Implementation". Could tighten slightly to say something like "and this would introduce a double summation in Eqn(1), over n locations. Hence R and B become matrices of size [n*s x n*s]?" Is that correct? This would fit with, as stated, to find "values for a common set of parameters". This gives single values for each z parameter. The wording of the last sentence is slightly ambiguous? "Similarly, the first and second derivative. . ...using the sum of the derivatives at the individual sites". This reads as if the derivatives are calculated locally, and then a mean taken. Would in fact a single sweep across all n*s data points be used, and the derivatives calculated once, if common parameters are investigated. [Maybe eqn term cancellation implies they are the same, but. . ...?].**

The following equation has been added to the section to clarify the methodology with $s$ denoting different sites:

$$f\left(\vec{z}; \hat{\vec{z}}, \vec{z}_0\right) = \frac{1}{2}\left[\sum_s \sum_t (\mathbf{m}_{t,s}(\vec{z}) - \mathbf{o}_{t,s})^T \mathbf{R_s}\left(\hat{\vec{z}}\right)^{-1}(\mathbf{m}_{t,s}(\vec{z}) - \mathbf{o}_{t,s}) + S\lambda(\vec{z} - \vec{z}_0)^T \mathbf{B}^{-1}(\vec{z} - \vec{z}_0)\right]. \quad (1)$$

**Somewhere in Section 2.3 or 2.4 ? possibly remind readers that FLUXNET also comes with the meteorological data. (In other words, it's not just the fluxes and then something like NCEP or ECMWF data was used additionally to give met drivers).**

Text added to P6 L16: *"... using the meteorological forcing data.."*

**Section 2.5.2. Would need to be confident that outlier points didn't do something odd in Eqn (4)? I guess the initial sweep of data ensures this is OK? (The alternative would be to normalise with SD of (mi,t)), and then get the percentage of variance explained).**

Thank you very much for this suggestion. By normalising with the standard deviation, we realised how similar our metric was to the fraction of variance unexplained. The fraction of variance unexplained is a useful metric used in statistical analysis and suits our problem well, therefore was picked has an alternative. The values are more intuitive, with 0 still representing a perfect match to the observations. The following text now replaces lines 22-27 on page 7:

*"For each data stream i, the fraction of variance unexplained by the model is*

$$\epsilon_i^2 = \frac{\sum_{t=1}^{k}(\vec{o}_{i,t} - \vec{m}_{i,t})^2}{\sum_{t=1}^{k}(\vec{o}_{i,t} - \vec{\bar{o}}_i)^2}, \quad \text{where} \quad \vec{\bar{o}}_i = \frac{1}{k}\sum_{t=1}^{k}\vec{o}_{i,t} \tag{2}$$

*It follows that the mean fraction of variance unexplained across data streams,*

$$\epsilon^2 = \frac{\epsilon_1^2 + \epsilon_2^2}{2}, \tag{3}$$

*is a single dimensionless measure of model misfit. The fractional error $\epsilon$ can than be interpreted as the typical (root-mean-square) error expressed as a fraction of the (root-mean-square) magnitude of the observed seasonal cycle. Thus, $\epsilon = 0$ represents a perfect match to the observations, while $\epsilon = 1$ corresponds to the error in a null model whose prediction $\vec{m}_{i,t}$ always equals the observational mean $\vec{\bar{o}}_i$. "*

**Figure 3. This is a great figure. However, pictures and captions often get pulled out of papers and shown in isolation. To ensure information is safely contained, mention in caption (or across top of plot), these are broadleaf trees only? At first glance, I thought the y-axis was some sort of physical unit (for LE or NPP). However caption says this is from the 2.5.2 Section metric. The normalised values are very small, and don't ever get near unity. Could this be the outliers mentioned above? Not a problem, but bottom of page 7 says "1 ? a complete mismatch" Wouldn't we expect some of the parameters to perform quite badly, and get a bit nearer to unity? Or - does this mean that in general, even without parameter fitting, then JULES is an exceptionally good model? Fitting reduces that last small error down further? Figure 3 mentions training versus validation. This appears different to the impression of the Abstract that all data is used to train?**

Section 3.2 was put in to asses the multisite methodology. It showed that given a set of 5 randomly selected sites, the optimised parameter vector found by optimising over these sites also improves the rest of the sites not used in the calibration. This experiment is now obsolete since we have the ability to validate the PFT-specific parameters properly in the improved result section. As a result, this figure has been removed. The altered metric, as described in the previous comment, is now more intuitive. A value 1 no longer represents a complete mismatch, which was unclear, but corresponds to the misfit of a null model whose prediction is equal to the mean observation at every time point.

**Figure 3. Usual practice is to put the legend inside the plot ? there is space for the 5 symbols, top left hand maybe?**

Though Figure 3 no longer exists (see response above), Figure 6 is very similar and the legend has been moved inside the plots for this figure.

**Figure 4 is great. But on my print out, the vertical lines cannot be seen in many instances. Thicken them maybe? As always, a matter of style, so just a suggestion. To make Figure 4 less crowded, would it make sense to not put the value of original & optimised as text annotations as this repeats information in the plots. Then the plots can be made bigger and bolder? Maybe put units in left column?**

These are nice ideas. In order to declutter this figure, the prior and posterior values have been removed since this values are made explicit in Table B1. This has allowed us to increase the error bar plots. The lines have been thickened to improve readability. Units have been left out in order not to re-clutter the figure.

**Section 3.3 and Figure 5. What is so remarkable about Figure 5 is that the strong correlations between parameters are not consistent across the PFTs. Maybe not for this paper, but some sort of physical interpretation of that would be really interesting. Returning to the governing equations and their scaled amounts might help. Is "correlation" the best word ? "collinearity" might be more appropriate?**

This section has been expanded to include a more thorough analysis of the correlations in the context of physical meaning. In order to achieve this, a description of the relevant JULES equations has been added to section 2.1. This puts the parameters in terms of the equations they govern. These equations are then used in section 3.2 to explain why some of the parameters vary the way they do.

**Figure 6 is like Figure 3, but a lot less cluttered. The data on Figure 6, BTs is same as that on Fig 3, except the "multi-site". Looking back at Fig 3, need to understand better the "five sites" algorithm (again, page 10, line 4 - some text accidently deleted?).**

As mentioned in response to the comment above Fig 3, the section containing Fig 3 has been suppressed and only Figure 6 remains.

**Figure 6 - I'd make the lower y-axis bound 0.0 (rather than what looks like 0.001)? Gives a better feel then of the improvement in absolute terms.**

With the amended metric, a log scale is now used in this figure. It follows that a fixed vertical improvement in this figure represents a fixed (multiplicative) reduction in fractional error.

**Conclusions - To my eye, Figure 6 says it all, and I would stress far more the real headline findings that: (!) There is a general reduction in error of around 50% (2) Possibly of more importance, using cross-PFT parameters, often get very similar improvements than local fits. This implies robust parameterisations independent of geography - which GCM modellers always like to see.**

The conclusion has been expanded to emphasise this point.

**2 Small things:**

**Maybe get the words "Data Assimilation" used a few times on the paper on page 1 / Abstract, so it gets picked up for anyone using that expression in an Internet search. (It?s an older terminology used for this sort of approach, but is still valid).**

Added the term *"data assimilation"* in the abstract.

**Abstract: Line 2, maybe mention that JULES is also used comprehensively as an impacts tool, sometimes forced with known climatologies and/or alternative GCMs in to the future. So it is not used just coupled to UK Met Office models.**

Text added to abstract: *"JULES is also extensively used offline as a land-surface scheme impacts tool, sometimes forced with known climatologies into the future."*

**Abstract - could "automatically differentiated" be expanded slightly to "automatically differentiated with respect to JULES parameters. . .. . ."**

Text added.

**Maybe line 25, page 1. To make topical post COP21, could add something like: "Any future decreased ability of the land surface to draw-down atmospheric CO2 could imply fewer "permissible emissions" in order to stay below key warming thresholds such as two degrees"**

Text added.

**Top page 2. Is there a process other than nitrogen cycling that can be mentioned ? preferably one that has been introduced in to the JULES model version used here?**

Canopy light interception has been added to the text as a process modelled in the version 2.2 of JULES: *"..or canopy light interception (Mercado et al., 2009)."*

**Sentence "Given the small spatial footprint. . ..". Maybe clarify why this gives overtuning? Presumably because it might be see some anomalous plants in the small footprint, and that are not representative of PFTs over a broader area?**

Text added to clarify this *"This over tuning may be due to the fact that a single site may not represent the full range of a PFT, given different tree types, ages and aboveground biomass found at each site. There may be some anomalous plants in the small footprint that are not representative of the PFTs over a broader area."*

**Bottom of page 3. Line 30. Could mention that "available observations" are about independent large-scale measurements (such as FLUXNET)? These are different to the specific process measurements used to calibrate the individual components tha are mentioned in line 27.**

Text added: *"...available observations, such as eddy covariance flux data."*

**Page 4, line 5. Possibly: "As used widely in weather forecasting, along with other disciplines".**

Added.

**Page 4, line 14. Maybe: " . . ..outputs at time 1 <= t <= s" (so defines $s$).**

Since $s$ is not used anywhere in the paper, it has been removed from the sum.

**Page 4, line 22. To anyone new to data assimilation, could say: "optimal vector. . ..minimizes the cost function (Eq. 1) via JULES model itself though m=m(z) (left terms) and directly via z in the right terms"**

The sentence has been left unchanged, instead the term '$\mathbf{m}(z)$' has been added to Eq. 1. Hopefully this addition clarifies that the model time-series part of the cost function changes with different iterations of $z$.

**Page 4, line 25. I can understand box constraints = upper or lower bounds. But what does "limited memory" refer to?**

Compared to the full BFGS algorithm which stores a full approximation to the inverse Hessian, the limited memory version will only store a few vectors to represent the approximation (Bryd et al.,1995). For optimisation problems with large numbers of variables, this linear memory requirement makes the L-BFGS the variant of choice. Texted added to line 25: *"... to use limited memory, for computational affordability, and box constraints..."*

**On the diagram, Figure 1, right-hand side, maybe word as "Hessian to give uncertainty bounds"**

Text added to picture.

**Page 5, line 14. Table A1 has a lot fewer than 500 entries, so quite a lot of data is rejected?**

Of the 500 FLUXNET sites, we were only able to obtain 160. Of those, 50% were rejected. As well as the criteria listed in this section, sites were also excluded based on number of years available (2 years minimum in order to perform a spin-up) and whether the sites were dominated by particular PFT (e.g. more than 50% coverage with the exception of C4 grasses). The crop sites were excluded since in the newer versions of JULES these are considered separately. These extra exclusions are described in section 2.4. This is made explicit in the paper using the following text: *"Data from 160 sites were made available for this study by M.Groenendijk."*

**Page 7, line 15. "One or two" refers to whether a plot is a standard plot, or a contour plot in two parameters?**

This was referring to both and has hopefully been clarify with the addition of the text below:

*"Consequently the optimal parameter values (which are modes of the full high dimensional distribution) may not coincide with modes of the one- and two-dimensional marginal distributions."*

**Page 8, line 5. The analysis here is more testing the concept of common parameters between sites, rather than testing the methodology?**

This sentence has been suppressed with the removal of section 3.2.

**Figure 2 is really nice. Just a few small things. Is there are reason black is also dashed? Style thing, but I've have maybe put as the individual panel titles "Broadleaf LE, Broadleaf GPP", etc. So across the top of the panels. And then the y-axes, put the units ? so W/m2 left panels etc. Then possibly not bothered with the labels (a)-(e)? Maybe make the lines with slightly larger line width? Inside each box, give the site ID as annotated text, as these are time-series for just single sites.**

The black line denotes the observations; each point represents a FLUXNET observation and the dashed part of connects them. The figure has been changed to include the rest of the suggestions and the text changed according.

**Page 10, line 4. Some text missing from sentence?**

No longer relevant with the removal of section 3.2.

**References:**

[revised manuscript text omitted]

**3.2 PFT-specific optimal parameter values**

~~The effect of parameter vectors *z* vectors on the overall model-data fit at each of the sites tested, using the metric described in section 2.5.2. Original default JULES parameters (∗), site-specific optimal parameters (∗), and the multisite parameters found by optimising over each set of five sites (,• ∗ 0∘,,• ∗ 0∘), denoted set 1, set 2, set 3, set 4 respectively. Sites in the training set (filled circles), sites in validation set (open circles).~~

~~The optimised parameter vectors generally perform well, both on the sites used in the training sets and the sites used in the validations sets. Indeed 15/18 of the sites improve no matter which of the optimised parameter vectors are used. The parameter vector optimised over set 3 performs even better than the individual optimisations for some of the sites. JULES performed worse on just two sites (UK-PL3, US-Ha1) using these parameter values compared to the default JULES parameters. These two sites also start off with relatively small errors, so even with the slight increase in errors they are still among the best performing sites in the set. UK-PL3 does not improve with any of the 5-site parameter sets, but observations from this site appears to be somewhat unusual (e.g. it has a very different seasonality to the rest of the sites for this PFT).~~

**3.3  PFT parameter values**

Optimisations were performed over all available sites for each of the PFTs simultaneously. The optimised model parameters for each of the PFTs are presented in Fig. 3.

For half of the parameters, the  prior parameter value lies outside the posterior uncertainty bounds. The $\frac{\delta c}{\delta l}$ parameter, which determines the efficiency of rainfall interception by the plant canopy,

does not change much from its original value for any of the PFTs. The uncertainty bounds are relatively tight and symmetrical.

30 The rest of the parameters show more variation. As described in section 2.5.1, the optimal values need not be in the centre of the uncertainty range, the  PDF can be skewed. Most of the PFTs display high uncertainty in at least one of the  parameters optimised; for the optimised broadleaf set for example, $dq_c$ is highly unconstrained. For C4 grasses, $d_r$ is so unconstrained  that the optimal value found  lies outside the 80% confidence interval. C3 grasses  show large uncertainty in $n_0$ and for shrubs, the parameter with the largest uncertainty is $\alpha$.

The uncertainties shown in Fig. 3 are one-dimensional marginal distributions. To understand further how the parameters are correlated, consider the two-dimensional representation in Fig. 4. For all of the PFTs, the  posterior parameter uncertainties

5 exclude a large part of the prior ranges. The cloud of plausible points tends to be restrictive and tight for most parameters.

~~The majority of the broadleaf parameters, shown in Fig. 4(a), are highly correlated with each other. The $d_r$ and $\frac{\delta c}{\delta l}$ parameters are the only ones to be uncorrelated with other parameters. Similarly, the needleleaf parameters (Fig. 4(b)) are all highly correlated, either positively or negatively, except for $T_{\text{low}}$, which is completely uncorrelated with any of the other 
[revised manuscript text omitted]

[Figure]

(a) Broadleaf

(b) Needleleaf

(c) C3 grasses

[Figure]

(d) C4 grasses

**Figure 4.** continued

[Figure]

(e) Shrubs

**Figure 4.** continued

[Figure]

The effect of different parameter vectors on the overall model-data fit at each of the site tested, using the metric described in section 2.5.2. The three z vectors tested: the original default JULES parameters (∗), the parameter vector found by optimising at the individual sites (∗), the new PFT-specific parameter vector found by optimising over the given PFT (●). Outliers with very large initial errors have been removed from the plot shown (Broadleaf: BR-Sa3, IT-Lec, Needleleaf: CA-NS2, SE-Sk2, IT-Yat, IT-Lav).

[revised manuscript text omitted]

---

## Referee Report (RR1)

**Review of "Land surface parameter optimisation through data assimilation: the adJULES system" by NM Raoult, TE Jupp, PM Cox and CM Luke, submitted to CMD**

**Reviewed by Professor Hoshin Gupta (The University of Arizona, Tucson, AZ, USA)**

**Objective of this paper:** To demonstrate application of automated local-search parameter optimization methodology to calibrate the parameters of the Joint UK Land Environment Simulator (JULES) land- surface model against eddy covariance measurements of gross primary production (GPP) and latent heat (LE) fluxes. The approach is used to define optimised parameter values (along with uncertainty estimates) for the 5 Plant Functional Types (PFTs) in JULES, improving the calibration and evaluation performance at 85% of the study sites used.

Data of GPP flux and LE flux from 160 sites were used. Input fields of vegetation structure and soil type were drawn from the UK Met Office files. LAI is derived from a MODIS product. The values taken for each site correspond to the closest grid point at which data are available.

**Overall Comments:** Overall I found the paper to be clear, well organized and well presented. As a report of the application of a specific gradient-based optimization method to parameter calibration of a specific LSM, the manuscript succeeds quite well. However, my assessment is that, as a general contribution to the scientific literature on the topic of LSM model performance improvement by application of data-assimilation for parameter estimation, the paper can certainly be improved.

For example the review of related literature [citing *Wang et al. (2001, 2007); Reichstein et al. (2003); Knorr and Kattge (2005); Raupach et al. (2005); Santaren et al. (2007); Thum et al. (2008); Williams et al. (2009); Peng et al. (2011), Xiao et al., (2011), Kuppel et al., (2012, 2014), Medvigy et al. (2009), Verbeeck et al. (2011), Medvigy and Moorcroft (2011), Groenendijk et al. (2010)*] fails to cite much of the related literature dating back to at least 1999 [e.g., *Gupta et al (JGR 1999), Houser et al (JGR 2001), Laplastrier et al (JGR 2002), Xia et al (JHM 2002), Demarty et al (JoH 2004), Liu et al (JGR 2004), Liu et al (JHM 2005), Demarty et al (WRR 2005), Hogue et al (WRR 2006), Abramowitz et al (JHM (2007), Rosolem et al (HP 2012)*, to name just a few].

While I certainly do not expect the authors to cite all of the above mentioned examples (which would be embarrassingly self-serving), I am generally concerned that the discussion does not show much awareness of the related developments in other closely related fields (such as Hydrologic Science) from which much of the impetus for optimization of the parameters of LSM's derives … beyond referring to the motivation being "*ideas from the applied mathematics of data assimilation as used widely in weather forecasting and other disciplines, and motivated by pioneering attempts at carbon cycle data assimilation (Rayner et al. (2005); Kaminski et al. (2013))*". Certainly, to my knowledge, much early work in LSM parameter optimization was promoted by the group led by Professor Pitman in Sydney. And as such, the general attention to parameter optimization considerably pre-dates the current interest in the broader concept of "*data assimilation*".

Further, I would generally expect a manuscript of this kind to provide a comparative evaluation of the model performance improvements with those of related parameter optimization studies performed by other members of the LSM community (albeit not using derivative based optimization).  And while I am not arguing that a comparison with other kinds of optimization methods necessarily needs to be performed (although it would be useful and informative) I do

think it would be prudent to provide some comments about a) computational cost and b) relative advantages and disadvantages vis-a-vis other optimization approaches that have been used for LSM parameter optimization by the community (beyond simply remarking that the method is prone to premature convergence at local optima).

Nonetheless, I do have some more specific and serious concerns that are worthy of attention as indicated below. Please understand that my comments are intended to be helpful (from the perspective of my experience with such issues) to the authors to improve their manuscript and in no way should be interpreted as a criticism of the nice work that has already been performed and reported here.

**Specific Comments:**

*Section on Methods and Data:*

1) **Qn:** Only eight parameters relating predominantly to leaf-level stomatal conductance and photosynthesis (including the hydrological partitioning at the land-surface) are calibrated. Please comment on the fact that given there are *"over a hundred internal parameters" in JULES that need to be specified, and that "the detailed performance of a land-surface model can be very sensitive to such internal parameters",* it is quite possible that fixing most of the parameters during the optimization might affect the calibration results (due to parameter interdependence effects). Further, the model likely contains additional coefficients that are fixed (hard-coded) to values that may be generally suspect (see *Mendoza et al WRR 2015*); please comment on potential the implications of that to the results obtained.

   Mendoza PA, MP Clark, M Barlage, B Rajagopalan, L Samaniego, G Abramowitz and H Gupta (2015), *Are we unnecessarily constraining the agility of complex process-based models?* Water Resources Research

2) **Qn:** While you point out the inherent "*subjectivity*" and lack of reproducibility of LSM parameter calibration by manual adjustment, please comment on the history (since at least 1999) of the application of "*objective*" automated (multiple-criteria) methods to LSM calibration, albeit not with gradient based algorithms, and also please comment on the relative strengths and weakness of manual versus automated methods. I should also point out that the mathematical basis for such automated calibration significantly pre-dates the "*data assimilation*" literature that the authors cite, and goes back at least as early as Bard (1974, Nonlinear Parameter Estimation, Academic Press) as a well established reference.

3) **Qn:** Please comment on the reliability of the "*parameter uncertainty estimates*" provided by adjoint methods (linear-Gaussian approximation), given the significant parameter-output nonlinearity associated with LSMs (I note, in particular, your comment that "*optimal values need not be in the centre of the uncertainty range, the PDF can be skewed*". For example, does empirical Monte-Carlo sampling of the region of the optimum (thereby approximating the true non-Gaussian shape of the posterior parameter pdf) provide similar uncertainty ranges for the parameters?

4) **Qn:** Continuing on from the above, it appears that for the results you did not actually use the Hessian generated by adJULES to report the uncertainty estimates for the parameters, but instead used sampling of the posterior distribution (Section 2.5.1),

which makes perfect sense. Since the earlier part of the paper gives one the impression that the uncertainty estimates are computed directly by adJULES it would be good to modify the presentation to make the actual fact clear (remove possibility for confusion).

5) **Qn:** Please comment on the fact that use of an additive cost function (where only the total summed cost is minimized, as opposed to an approach where all individual cost functions are required to be improved) means that it is possible for the optimization method to achieve the "*best*" solution by improving the match at one site while possibly making the match worse (than it could otherwise be) at one or more other sites simply to achieve a better value of the cost function.

6) **Qn:** Please comment on the fact that, due to model structural errors, calibration to specific observables could actually cause model simulations of other fluxes (that were not used in tuning) to become worse (this is a potentially very serious problem in multi-flux calibration by weighted single criteria optimization to only some of the model fluxes); please see *Gupta et al (JGR 1999)*.

Gupta HV, L Bastidas, S Sorooshian, WJ Shuttleworth and ZL Yang (1999), Parameter Estimation of a Land Surface Scheme Using Multi-Criteria Methods, GCIP II Special Issue of the Journal of Geophysical Research-Atmospheres, Vol. 104, No. D16, p. 19491-19503

7) **Qn:** I am concerned that the $\epsilon_i^2$ "*fractional variance explained*" measure is not really a properly informative measure of model performance, given that the benchmark for comparison is the observed seasonal cycle (I note also the related comment by a previous reviewer). In Hydrology this is related to a metric known as the "*Nash*" efficiency (equivalent to $1 - \epsilon_i^2$) that has been repeatedly demonstrated to be a poor index of model performance unless $1 - \epsilon_i^2 > 0.85$ or $0.9$ (such values are rarely achievable), and can hide the existence of significant bias in the performance (see, e.g., *Schaefli and Gupta (HP 2007)*). Note that the "Nash" efficiency is also typically justified as enabling cross-site and cross-model comparison, but arguable this is a poor reason for using a poorly informative metric. Instead the component decomposition [e.g., see *Murphy (Monthly Weather Review 1988)*, *Gupta et al (Journal of Hydrology 2009)*] can provide a more meaningful indicator of performance in terms of bias, variability and cross-correlation (see application in Rosolem et al, Hydrological Processes 2012).

Murphy A (1988), Skill Scores based on the Mean Square Error and their Relationships to the Correlation Coefficient, Monthly Weather Review 116, 2417-2424.

Rosolem R, HV Gupta, WJ Shuttleworth, LGG de Goncalves, and X Zeng (2012), Towards a Comprehensive Approach to Parameter Estimation in Land Surface Parameterization Schemes, Hydrological Processes, published online in Wiley Online Library (wileyonlinelibrary.com) DOI: 10.1002/hyp.9362

Gupta HV, H Kling, KK Yilmaz and GF Martinez-Baquero (2009), Decomposition of the Mean Squared Error & NSE Performance Criteria: Implications for Improving Hydrological Modelling, Journal of Hydrology, Vol. 377, pp. 80-91, doi: 10.1016/j.jhydrol.2009.08.003

Schaefli B and HV Gupta (2007), Do Nash values have value?, Hydrological Processes, 21(15), 2075-2080, simultaneously published online as Invited Commentary in Hydrologic Processes (HP Today), Wiley InterScience, doi: 10.1002/hyp.6825

**Section on Results and discussion:**

8) **Qn:** Please comment about your results in the context of the findings by *Abramowitz et al (JHM 2007)* who show that "*… as much as 45% of per-time-step model root-mean-*

*square error in … flux outputs is due to systematic problems in … model processes insensitive to changes in vegetation parameters … These results suggest that efforts to improve the representation of fundamental processes in land surface models, rather than parameter optimization, are the key to the development of land surface model ability"*.

Abramowitz G, A Pitman, HV Gupta, E Kowalczyk and Y Wang (2007), Systematic Bias in Land Surface Models, Journal of Hydrometeorology, 8(5) pp 989-1001

9) **Qn:** Please comment on the sensitivity of the optimized results to choice of the constant of proportionality λ (how do the results change if λ is made smaller). Given the importance of using a "prior" on the parameters as constraint, this seems to me to be a rather important issue.

10) **Qn:** Please comment on the quality of the site-specific performance improvement in comparison with findings obtained by others. For example, *Rosolem et al (Hydrological Processes 2012)* reported that "*All sites showed improvements in simulation of the surface energy and carbon fluxes*" and "*In contrast, the default parameter sets (commonly used in GCM simulations) were found to be unable to reproduce the diurnal variation of energy fluxes at the tropical rainforest sites and showed a tendency to overestimate (underestimate) sensible (latent) heat fluxes. The calibration improved the simulations of these two fluxes by removing bias and variability errors (errors in signal mean and standard deviation).*"

11) **Qn:** In general it is well accepted that it is not ever possible to "*validate*" a model, and so the term validation in regards to model performance evaluation would seem to be misleading. Might I politely suggest the use of the more accurate term "*evaluation*" in its stead?

Section on Conclusions:

12) **Qn:** Perhaps it would be appropriate to comment on the computational cost involved with optimization using the adJULES system and on how many model runs (cost function evaluations) are necessary to achieve convergence (starting from the default parameter values)?

---

## Author Response (AR2)

**Response to Reviewer #3 (our responses in italics)**

Ref #3: This paper used the adjoint technology, L-BFGS-B optimization algorithm and Gibbs sampler to optimize the parameters of JULES. Generally speaking, this paper is well written since it has been reviewed and revised, but it also has three problems that I have to point out. (1) Data assimilation. This paper used 'adjoint' to provide the gradient information. Although 'adjoint' has been used in data assimilation, it is actually not 'data assimilation'. So the 'data assimilation' in the title, as well as in the main body, are suggested to be replaced by 'adjoint'. (2) Local optimum. If I understand correctly, the gradient based BFGS algorithm is used to find the local optimum, but if there are multiple local optimums, how to jump out of the local optimums to find the global one? (3) Posterior distribution. The posterior distribution seems really weird and I guess the reason is a too strong assumption: truncated multivariate normal distribution. If the true posterior distribution is not Gaussian, the results given by Gibbs sampler with such a strong assumption might be very misleading.

**(1)      Title: data assimilation**
What's the definition of 'data assimilation' in your paper? For my all due respect, this title is really misleading. Usually, data assimilation means adjusting the state variables, such as soil moisture, temperature, and pressure, with observed values; while parameter optimization means tuning the parameters according to the deviation between observed and simulated output. Although they have some common techniques, these two concepts are fundamentally different. Kalman filter, 3D-Var and 4D-Var are very popular in the community of data assimilation, while in the community of parameter optimization, Genetic Algorithms, Metropolis-Hastings, and Gibbs samplers et.al., are more frequently used. Consequently, it is really weird to use 'optimization through data assimilation' in the title of this paper, because the adjoint (Hessian) of JULES, BFGS gradient based optimization and Gibbs sampler, were used to tune the parameters, while the state variables remain untouched. Although the 'adjoint' technique has been used in data assimilation, it is not equals to data assimilation. I think it might be better to replace 'data assimilation' with 'adjoint'.

*__Our Response:__ We acknowledge that we use a broader definition for "Data Assimilation" than is often the case in weather forecasting. Our application of data assimilation involves using data assimilation techniques to estimate internal model parameters, while DA in weather forecasting is typically used to estimate the initial state of the atmosphere. However, we are certainly not the first to use "Data Assimilation" in this sense (see for example Rayner et al., 2005). In addition Reviewer #1 has previously requested that we use the term "Data Assimilation" more rather than less prominently (in contradiction to the suggestion of Reviewer #3). In order to avoid a potential impasse, we have therefore retained the term but explained our use of it more clearly by the addition of the following text at the beginning of the data assimilation section.*

*__"The term 'Data Assimilation' is commonly used to describe the process of using observations to refine the initial state within a numerical representation of a system  (Bouttier and Courtier, 1999).  This is most obviously the case for weather forecasting, in which the temperature, humidity and wind fields define the initial state. However, data assimilation techniques have been also used for parameter estimation, for example in hydrological models (Madsen, 2003, Liu and Gupta, 2007), and carbon cycle data assimilation systems (CCDAS, (Rayner et al. (2005); Kaminski et al. (2013)). In parameter optimisation by data assimilation, the internal parameters of a model take on the role of the dynamical state variables in initial state estimation by data assimilation. Nevertheless, the underlying techniques (e.g. of defining a model adjoint and minimising the error in the fit to data), are very similar in these two applications of data assimilation. This paper is certainly not the first to define parameter estimation of this form as data assimilation (Braswell et al. (2005), Stockli et al (2008), Verbeeck et al. (2011), Kuppel et al (2012), Hararuk et al. (2014)), but the reader should note the subtle difference between our definition of data assimilation and that commonly used in weather forecasting."*

Ref #3:  (2)    Page 2, line 5: two major sources: (a) process uncertainty, and (b) parameter uncertainty…

You are missing another important source, the uncertainty due to initial and boundary conditions (forcing data, for land surface models). [Kavetski et al., 2006a, 2006b][Ajami et al., 2007]

*Our Response: Thanks, this is a good point. We have included the following relevant text:* **"Uncertainties in LSMs arise from three major sources: parameter uncertainty, process uncertainty and uncertainty due to initial and boundary conditions. Taking these in reverse order, uncertainty due to initial and boundary conditions in the case of LSM refers to uncertainty in the forcing data (Kavetski et al. (2006a, 2006b), Ajami et al.( 2007))."**

**(3)    Page 3, line 15, Page 5, line 1 and 12, etc.**
Replace 'data assimilation' with 'Adjoint based optimization' or something that can accurately describe the methodology used in this paper. See my previous comment (1).

*Our Response: See response to comment (1).*

**Ref #3:  (4)    Page 6, line 12**: gradient descent algorithm L-BFGS-B, **Page 6, line 22**: locally optimal parameter vector, **Page 6, lind 24**: locally optimized parameters

This paper used the gradient descent algorithm L-BFGS-B as optimization method, and the gradient information was provided by 'adjoint'. My concern is, if the optimization algorithm is gradient base, how to deal with the local optimums? If the response surface has multiple local optimums, the gradient-based methods will be trapped in a local optimum so that they cannot find the global optimum. There have been a lot of approaches to find the global optimum, such as Genetic Algorithm [Wang, 1991], SCE-UA [Duan et al., 1992], or multi-start methods [Krityakierne and Shoemaker, 2015]. Although the performance of JULES has been significantly improved, it is still not perfect for me because the model performance can be further improved if the algorithm can find the global optimum. Please provide convincing information that the algorithm can successfully jump out of local optimums, or use another algorithm to find the global optimum.

*Our Response: Getting stuck in local optima is indeed a potential shortcoming of any gradient-descent methodology. However, the consistency between our single site and multiple site optimisations gives us confidence in the robustness of the convergence of our algorithm for this application. The gradient-descent methodology, albeit with the slower tangent linear method, has also been used in other LSM parameter estimation papers e.g. Kuppel et al. 2012. However, we have added text to acknowledge this issue:*

**"There is always a risk of becoming stuck in local minima when optimising within a high- dimensional parameter space by gradient descent. When an optimisation finds a local minimum, the final optimised state depends on the initial conditions. The consistency between our single site and multiple site optimisations therefore gives us some confidence in the robustness of the convergence of our algorithm for this application ."**

**Ref #3 : (5)    Page 9, line 17**: a truncated multivariate normal distribution … using Gibbs sampling…
This is actually a very strong assumption about the posterior distribution. If you want to obtain the posterior distribution of parameters but do not know too much about the shape of it, more general approaches, such as Metropolis-Hastings[Hastings, 1970], Adaptive metropolis[Haario et al., 2001], or DRAM[Haario et al., 2006], might be better.

I think the reason of strongly correlated joint posterior distribution is your strong assumption of truncated multivariate normal distribution. If you assumed the posterior distribution to be multivariate normal, it is impossible to get any other shape of distribution. Due to my experience, the response surfaces of land surface models are usually not Gaussian, i.e. have multiple local optimums, valleys and peaks, so that such a strong assumption will give totally misleading results. So I suggest to use more general MCMC approaches, such as MH, AM or DRAM to replace Gibbs.
*Our Response: We assume uniform top-hat priors on the initial parameters given their upper and lower bounds. This leads to truncated Gaussian posteriors. Since the posterior distribution is truncated, we cannot simply extract a simple probability density function. At*

*the optimum found using the gradient descent method, the Hessian is used to approximate the local quadratic around the optimum. Gibbs sampling is used to sample this space. This can be done without requiring further model runs.*

**Ref #3 : (6)    Page 14. Section 3.2.1** should be section 4.

***Our Response:*** *Section 3 is titled 'Results and discussion' and section 3.2.1 'Assessment of PFT-specific optimal parameters'. Whilst we agree that putting this part as a subsubsection detracts from its importance, it does fit well in the section 3 description. As a compromise, section 3.2.1 has been moved up a level to be a subsection of section 3.*

**Ref #3 : (7)    Page 17, line 10:** Alternative methods … could avoid this issue, but are more computationally costly.

How many model evaluations did it cost in this paper? Some classical global optimization methods, such as SCE-UA [Duan et al., 1994], are able to converge within several thousands of model evaluations, while the most recent surrogate based optimization algorithms, such as [Wang et al., 2014; Gong et al., 2015, 2016], are able to obtain global optimum with only hundreds of model evaluations. The global optimization methods are not as costly as you think.

***Our Response:*** *For this paper, optimisations typically took 150 function evaluations. Whilst we acknowledge that there are other methods, this does remain one of the most effective, and the purpose of this study was to explore what could be found using the model adjoint.*

**Response to Reviewer #4: Professor Hoshin Gupta (our responses in italics)**

**General Comments:**

Ref #4: For example the review of related literature [citing *Wang et al. (2001, 2007); Reichstein et al. (2003); Knorr and Kattge (2005); Raupach et al. (2005); Santaren et al. (2007); Thum et al. (2008); Williams et al. (2009); Peng et al. (2011), Xiao et al., (2011), Kuppel et al., (2012, 2014), Medvigy et al. (2009), Verbeeck et al. (2011), Medvigy and Moorcroft (2011), Groenendijk et al. (2010)*] fails to cite much of the related literature dating back to at least 1999 [e.g., *Gupta et al (JGR 1999), Houser et al (JGR 2001), Laplastrier et al (JGR 2002), Xia et al (JHM 2002), Demarty et al (JoH 2004), Liu et al (JGR 2004), Liu et al (JHM 2005), Demarty et al (WRR 2005), Hogue et al (WRR 2006), Abramowitz et al (JHM (2007), Rosolem et al (HP 2012)*, to name just a few].

While I certainly do not expect the authors to cite all of the above mentioned examples (which would be embarrassingly self-serving), I am generally concerned that the discussion does not show much awareness of the related developments in other closely related fields (such as Hydrologic Science) from which much of the impetus for optimization of the parameters of LSM's derives ... beyond referring to the motivation being "*ideas from the applied mathematics of data assimilation as used widely in weather forecasting and other disciplines, and motivated by pioneering attempts at carbon cycle data assimilation (Rayner et al. (2005); Kaminski et al. (2013))*". Certainly, to my knowledge, much early work in LSM parameter optimization was promoted by the group led by Professor Pitman in Sydney. And as such, the general attention to parameter optimization considerably pre-dates the current interest in the broader concept of "*data assimilation*".

***Our Response:*** *We thank you for this extensive insight into the Hydrological literature that will be invaluable. A reference to Pitman already exists at the beginning of the paper but we have now included a few of the other references listed above.*

Ref #4:  Further, I would generally expect a manuscript of this kind to provide a comparative evaluation of the *model performance improvements* with those of related parameter optimization studies performed by other members of the LSM community (albeit not using derivative based optimization). And while I am not arguing that a comparison with other kinds of *optimization methods* necessarily needs to be performed (although it would be useful and informative) I do think it would be prudent to provide some comments about a) computational cost and b) relative advantages and disadvantages vis-a-vis other optimization approaches that have been used for LSM parameter optimization by the community (beyond simply remarking that the method is prone to premature convergence at local optima).

***Our Response:***  *As far as we are aware there are no other such studies performed on the JULES land-surface model. The Blyth et al. benchmarking study outlines the performance of JULES using the default parameters and this study improves the fit at the sites used in the benchmarking amongst many more sites. Computational cost information has now been added to the main text.*

**Section on Methods and Data:**

**Ref #4: (1)** Only eight parameters relating predominantly to leaf-level stomatal conductance and photosynthesis (including the hydrological partitioning at the land-surface) are calibrated. Please comment on the fact that given there are "*over a hundred internal parameters*" in JULES that need to be specified, and that "*the detailed performance of a land-surface model can be very sensitive to such internal parameters*", it is quite possible that fixing most of the parameters during the optimization might affect the calibration results (due to parameter interdependence effects). Further, the model likely contains additional coefficients that are fixed (hard-coded) to values that may be generally suspect (see *Mendoza et al WRR 2015*); please comment on potential the implications of that to the results obtained.

***Our Response:*** *The following sentence has been added to the end of line 25 in section 2.4:*

**"*Note that this only represents a modest subset of the parameters available and as such the results could be different when considering different subsets in the calibration."***

**Ref #4: (2)** While you point out the inherent "*subjectivity*" and lack of reproducibility of LSM parameter calibration by manual adjustment, please comment on the history (since at least 1999) of the application of "*objective*" automated (multiple-criteria) methods to LSM calibration, albeit not with gradient based algorithms, and also please comment on the relative strengths and weakness of manual versus automated methods. I should also point out that the mathematical basis for such automated calibration significantly pre-dates the "*data assimilation*" literature that the authors cite, and goes back at least as early as Bard (1974, Nonlinear Parameter Estimation, Academic Press) as a well established reference.

***Our Response****: We are of the view that our manuscript gives sufficient reference to other methods, within the confines of the problem that we address.*

**Ref #4: (3)** Please comment on the reliability of the "*parameter uncertainty estimates*" provided by adjoint methods (linear-Gaussian approximation), given the significant parameter-output nonlinearity associated with LSMs (I note, in particular, your comment that "*optimal values need not be in the centre of the uncertainty range, the PDF can be skewed*". For example, does empirical Monte-Carlo sampling of the region of the optimum (thereby approximating the true non-Gaussian shape of the posterior parameter pdf) provide similar uncertainty ranges for the parameters?

***Our Response:*** *We assume uniform top-hat priors on the initial parameters given their upper and lower bounds. This leads to truncated Gaussian posteriors. Since the posterior distribution is truncated, we cannot simply extract a simple probability density function. At the optimum found using the gradient descent method, the Hessian is used to approximate the local quadratic around the optimum. Gibbs sampling to used to sample this space. This is done without running any more JULES runs. The optimal values not lying at the centre of the PDF is due to the truncated and multidimensional nature of the posterior distributions.*

**Ref #4: (4)** Continuing on from the above, it appears that for the results you did not actually use the Hessian generated by adJULES to report the uncertainty estimates for the parameters, but instead used sampling of the posterior distribution (Section 2.5.1), which makes perfect sense. Since the earlier part of the paper gives one the impression that the uncertainty estimates are computed directly by adJULES it would be good to modify the presentation to make the actual fact clear (remove possibility for confusion).

***Our Response:*** *The Hessian is used to generate the uncertainty estimates, see response to comment (3) for more details.*

**Ref #4: (5)** Please comment on the fact that use of an additive cost function (where only the total summed cost is minimized, as opposed to an approach where all individual cost functions are required to be improved) means that it is possible for the optimization method to achieve the "*best*" solution by improving the match at one site while possibly making the match worse (than it could otherwise be) at one or more other sites simply to achieve a better value of the cost function.

***Our Response:*** *The following sentences have been added to the end of the Multisite Implementation section:*

**"*An additive cost function, where the optimisation criterion is to minimise the total cost, was chosen over a cost function where all individual cost functions are required to be improve. This was due to the fact that all of the sites were used finding the optimal parameter vector for each PFT. Sites which do not improve with the rest of the PFT will suggest wrong classification of the site or issues with the PFT definitions."***

**Ref #4: (6)** Please comment on the fact that, due to model structural errors, calibration to specific observables could actually cause model simulations of other fluxes (that were not used in tuning) to become worse (this is a potentially very serious problem in multi-flux calibration by weighted single criteria optimization to only some of the model fluxes); please see *Gupta et al (JGR 1999)*.

***Our Response:*** *The following sentence has been added to page 7 line 13:*

**"Also note that due to model structural errors, calibration to these two specific observables could actually cause model simulations of other fluxes (that were not used in tuning) to become worse (Gupta et al. 1999)."**

**Ref #4: (7)** I am concerned that the $\varepsilon^2$ "*fractional variance explained*" measure is not really a properly informative measure of model performance, given that the benchmark for comparison is the observed seasonal cycle (I note also the related comment by a previous reviewer). In Hydrology this is related to a metric known as the "*Nash*" efficiency (equivalent to $1 - \varepsilon^2$) that has been repeatedly demonstrated to be a poor index of model performance unless $1 - \varepsilon^2 > 0.85$ or $0.9$ (such values are rarely achievable), and can hide the existence of significant bias in the performance (see, e.g., *Schaefli and Gupta (HP 2007)*). Note that the "Nash" efficiency is also typically justified as enabling cross-site and cross-model comparison, but arguable this is a poor reason for using a poorly informative metric. Instead the component decomposition [e.g., see *Murphy (Monthly Weather Review 1988)*, *Gupta et al (Journal of Hydrology 2009)*] can provide a more meaningful indicator of performance in terms of bias, variability and cross-correlation (see application in Rosolem et al, Hydrological Processes 2012).

***Our Response:*** Since our cost function is described using RMSE, we felt the natural choice was to use a normalised version to perform the main diagnostic, especially in terms of cross-site comparison. However, we do agree that further breakdown of the metric would be beneficial. As such, we have decided to use Taylor diagrams since it provides a visual methodology to decompose the misfit into standard deviation and correlation.

A sentence has been added to end of section 2.5.2 in order to acknowledge this metric's shortcomings and section 3.4 has been included displaying a few Taylor diagrams and their assessment.

*"In Hydrology this is related to a metric known as the "Nash-Sutcliffe" efficiency (Nash and Sutcliffe, 1970), equivalent to $1 - \varepsilon^2$, and has been used by many studies to perform cross-site comparisons."*

**Section on Results and discussion:**

**Ref #4: (8)** Please comment about your results in the context of the findings by *Abramowitz et al (JHM 2007)* who show that "*... as much as 45% of per-time-step model root-mean-square error in ... flux outputs is due to systematic problems in ... model processes insensitive to changes in vegetation parameters ... These results suggest that efforts to improve the representation of fundamental processes in land surface models, rather than parameter optimization, are the key to the development of land surface model ability*".

***Our Response:*** *We believe that this stated already in our conclusion on lines 10-14.*

**Ref #4: (9)** Please comment on the sensitivity of the optimized results to choice of the constant of proportionality λ (how do the results change if λ is made smaller). Given the importance of using a "prior" on the parameters as constraint, this seems to me to be a rather important issue.
limitation?

***Our Response:*** *The following sentence has been added to the end of section 3.2:*

*"**Choice of λ had less effect on the values of the optimal parameters than on uncertainties and correlations found. The uncertainty ranges become larger with smaller values λ and correlations less pronounced.**"*

**Ref #4: (10)** Please comment on the quality of the site-specific performance improvement in comparison with findings obtained by others. For example, *Rosolem et al (Hydrological Processes 2012)* reported that "*All sites showed improvements in simulation of the surface energy and carbon fluxes*" and "*In contrast, the default parameter sets (commonly used in GCM simulations) were found to be unable to reproduce the diurnal variation of energy fluxes at the tropical rainforest sites and showed a tendency to overestimate (underestimate) sensible (latent) heat fluxes. The calibration improved the simulations of these two fluxes by removing bias and variability errors (errors in signal mean and standard deviation)*."

***Our Response:*** *It is not clear how we can compare diurnal cycles in Rosolem et al to the annual cycle calibrations we perform in our study.*

**Ref #4: (11)** In general it is well accepted that it is not ever possible to "*validate*" a model, and so the term validation in regards to model performance evaluation would seem to be misleading. Might I politely suggest the use of the more accurate term "*evaluation*" in its stead?

***Our Response:*** *The use of the word validation is due to its established pairing with the word calibration. Many studies use calibration and validation, and whilst substitution would not change the science, we prefer to stick with the word 'validation'.*

**Section on Conclusions:**

**Ref #4: (12)** Perhaps it would be appropriate to comment on the computational cost involved with optimization using the adJULES system and on how many model runs (cost function evaluations) are necessary to achieve convergence (starting from the default parameter values)?

***Our Response:*** *Text to this effect has been added to end of the first sentence in section 3.1: '...typically requiring 150 function evaluations to find a local optimum.'*

**REFERENCES:**

[revised manuscript text omitted]